# Augmented Memory Replay-based Continual Learning Approaches for Network Intrusion Detection

**Suresh Kumar Amalapuram**∗, **Sumohana S Channappayya**#, **and Bheemarjuna Reddy Tamma**∗
Dept. of Computer Science and Engineering∗, Dept. of Electrical Engineering#
Indian Institute of Technology Hyderabad, India
`{cs19resch11001,sumohana,tbr}@iith.ac.in`

## Abstract

Intrusion detection is a form of anomalous activity detection in communication network traffic. Continual learning (CL) approaches to the intrusion detection task accumulate old knowledge while adapting to the latest threat knowledge. Previous works have shown the effectiveness of memory replay-based CL approaches for this task. In this work, we present two novel contributions to improve the performance of CL-based network intrusion detection in the context of class imbalance and scalability. First, we extend class balancing reservoir sampling (CBRS), a memory-based CL method, to address the problems of severe class imbalance for large datasets. Second, we propose a novel approach titled perturbation assistance for parameter approximation (PAPA) based on the Gaussian mixture model to reduce the number of *virtual stochastic gradient descent (SGD) parameter* computations needed to discover maximally interfering samples for CL. We demonstrate that the proposed approaches perform remarkably better than the baselines on standard intrusion detection benchmarks created over shorter periods (KDDCUP'99, NSL-KDD, CICIDS-2017/2018, UNSW-NB15, and CTU-13) and a longer period with distribution shift (AnoShift). We also validated proposed approaches on standard continual learning benchmarks (SVHN, CIFAR-10/100, and CLEAR-10/100) and anomaly detection benchmarks (SMAP, SMD, and MSL). Further, the proposed PAPA approach significantly lowers the number of virtual SGD update operations, thus resulting in training time savings in the range of 12 to 40% compared to the maximally interfered samples retrieval algorithm.

## 1 Introduction

Learning from a continuum of data with distribution shift over time is known as continual/lifelong learning (CL) [1, 2]. CL algorithms aim to preserve learned knowledge (stability) while embracing the knowledge of new tasks (plasticity). The conflict in balancing stability and plasticity often leads to the problem of catastrophic forgetting (CF) [3, 4]. In other words, past tasks' knowledge degrades over time due to interference from the newer task. Unlike computer vision [5, 6, 7, 8], natural language processing [9, 10], and speech processing [11, 12], CL applications to many real-world problems are under-explored, and one such exemplar is network intrusion detection (NID) in communication networks. Intrusion detection is a form of distribution shift detection that aims to detect anomalous activity patterns in the network traffic [13, 14]. To ensure protection against novel cyber-attacks, network intrusion detection systems (NIDS) must evolve continuously and effortlessly learn from the newer attack data.

The concept drift/distribution shift is prevalent in cybersecurity, as malicious behavior can shift either suddenly or gradually over time [15, 13]. While NID formulated as anomaly detection (AD) is typically trained on normal data (*zero positive* learning [16]) and is immune to the drift of malicious

behavior, however, its performance can be severely affected when the distribution of normality shifts [14] (*open-world* setting). In the cybersecurity domain, a shift in normality can occur when new patches, software updates, devices, or protocols are introduced [14, 13]. This work considers normality shifts and distribution shifts in attack data. So, we formulate our problem as a continual learning-based supervised binary classification problem (SBCP).

**Class imbalance and Task-free nature**: SBCP formulation differs from the traditional classification problem as the former one possesses two challenging characteristics [17],i.e., *class imbalance* (CI) and *class overlap*. In this work, our focus is primarily on mitigating the adverse effects of class imbalance on performance. Here, CI means a *different number of training examples* ($n_t$) for each class in the dataset. The majority class contains the highest number of training examples, whereas minority classes are the last few classes when sorted in non-increasing order based on $n_t$. CI is said to be *severe* when the difference between the minority class training examples ($n_t$) is relatively high (e.g., the difference between the number of samples of DDoS attack-HOIC and SQL Injection attack (minority) classes of the CICIDS-2018 [18] dataset is 2.1 million). Intrusion detection generally operates in a streaming environment. As a result, task-free continual learning [19] becomes a natural fit for such applications. Working in a task-free CL setup assumes no information about task boundaries and the number of samples per class.

**Continual learning with shallow methods:** We study the feasibility of shallow methods (non-neural networks) in the CL setting by conducting preliminary experimenting using the *random forest* (RF) algorithm (a popular algorithm in NIDS literature) on CICIDS-2017 and CICIDS-2018 datasets. We made the following observations: First, RF exhibits higher CF (especially on minority classes) on previous tasks while learning new tasks. Second, augmenting RF with buffer memory partially reduces the CF on previous tasks. More details on this experiment are available in the supplementary material (**SM:A.2**). To conclude, compared to deep learning (neural network) methods, shallow methods exhibit higher CF, thus diminishing their performance in the CL setting.

Prior art [20] has demonstrated the suitability of CL for addressing the NID problem. This work shown the adverse impact of severe class imbalance on the performance of NID algorithms. This work also demonstrated the usefulness of memory replay (MR)-based approaches for this task. Generally, class imbalance-aware MR methods (e.g., class balancing reservoir sampling (CBRS) [21]) perform a periodic memory population operation to handle class imbalance. In particular, CBRS distinguishes majority and minority classes based on the respective class sample count (local information) from the buffer memory. However, this strategy has a pitfall when the count of minority class samples in the finite buffer memory appears to come from a majority class. This situation could occur when there is a severe imbalance among the minority classes, especially in larger datasets. Larger datasets mean the number of training samples in the order of millions (CICIDS-2018, AnoShift). In this work, we extend the CBRS method (and call it ECBRS) to address this issue. The proposed ECBRS always prioritizes the majority class for sample replacement using additional global information about class imbalance.

Table 1: Timing comparison between the virtual SGD operations and the total training of the MIR algorithm. Each experiment is repeated five times, and mean values are reported for virtual SGD ops time and total train time.

| Dataset | MIR (time in seconds) | | |
| --- | --- | --- | --- |
| | VSP ops time | Total train time | VSP ops proportion |
| CICIDS-2017 [22] | 102.9 | 254.3 | 40.4% |
| UNSW-NB15 [23] | 132.4 | 428.2 | 30.9 % |
| CTU-13 [24] | 142.4 | 398.1 | 35.7% |
| KDDCUP'99 [25] | 191.0 | 420.8 | 45.3% |
| AnoShift [26] | 499.4 | 1210.4 | 41.2% |
| CICIDS-2018 [22] | 3315.0 | 7620 | 43.5% |
| SVHN [27] | 112.0 | 217.0 | 51.6% |
| CIFAR-10 [28] | 58.6 | 118.6 | 49.4% |
| CIFAR-100 [28] | 45.71 | 88.56 | 51.61% |
| CLEAR-10 [29] | 128.2 | 265.9 | 48.2% |
| CLEAR-100 [29] | 703.6 | 1490 | 47.2% |

Other families of MR techniques like maximally interfered retrieval (MIR) [30] and gradient-based memory editing (GMED) [31] quantify the significance of samples in the buffer memory using temporary (virtual) stochastic gradient descent (SGD) parameter updates. However, this computation overhead consumes nearly 40 to 50% of the total training time, as shown in Table 1. Motivated by this drawback, we propose a novel approach known as perturbation assistance for parameter approximation (PAPA) for estimating the virtual SGD parameter (VSP) updates using the Gaussian mixture model (GMM) with low overhead.

To summarise, our key contributions in this work are:

1. An extension to the CBRS, dubbed ECBRS, to deal with severe class imbalance by always undermining the majority class by maintaining the global information about class imbalance. Further, ECBRS can be used as a memory population policy in conjunction with the existing memory replay-based approaches.

2. A perturbation assistance for parameter approximation (PAPA) method for estimating VSP updates that significantly reduces the training time for methods like MIR and GMED, leading to improved scalability.

3. A demonstration of the improved performance of the proposed methods relative to the considered baselines using standard network intrusion detection, computer vision, and anomaly detection benchmarks (including AnoShift and CLEAR-100, whose training data spans over ten years). Further, we show that using the PAPA method results in training time savings of 12 to 40% (relative to the baselines), thus it is highly suitable for large-scale training.

## 2 Related Work

**Intrusion detection:** An intrusion detection system (IDS) [32, 33] is one of the fundamental tools in security infrastructure to safeguard against cyber-attacks at the host or network (NIDS) perimeter level [34]. A signature-based NIDS uses a rule-based engine to identify known patterns, whereas anomaly-based NIDS identifies patterns that deviate from normal behavior [35]. NIDS are built using techniques ranging from traditional machine learning to deep learning and reinforcement learning [36]. Typically, network intrusions represent a tiny subset of network traffic [37]. As a result, a heavy class imbalance is observed in NID datasets, making the intrusion detection performance vulnerable. Sampling (algorithm) based approaches have been proposed in the literature to mitigate this problem [38, 39, 40]. But, this work focuses on building an NIDS that adapts to both benign and attack data distribution shifts without any aid of sampling strategies to handle the CI effect.

**Continual learning:** Recently, there has been a significant effort to expand the horizon of continual learning by mitigating the impact of CF [4]. These efforts can be broadly grouped into the following families [3]: regularization-based [41, 42], expansion-based [43], meta-learning-based [44], and rehearsal or memory replay-based approaches.

A subset of rehearsal methods that focus on class imbalance are CBRS [21] and partition reservoir sampling [45], in which samples are preserved in memory in proportion to their class strength. However, these methods do not handle severe CI between the minority samples. The proposed ECBRS addresses this CI issue. Further works like maximum interfered retrieval (MIR) [30] and gradient-based sample editing (GMED) [31] use virtual SGD parameter updates to identify the most interfered samples. These are samples in the replay memory that suffer an increase in loss due to previous parameter updates. However, for large-scale training, these virtual update computations add significant overhead. Unlike these, our proposed PAPA method significantly reduces this overhead, making it suitable for training over large datasets.

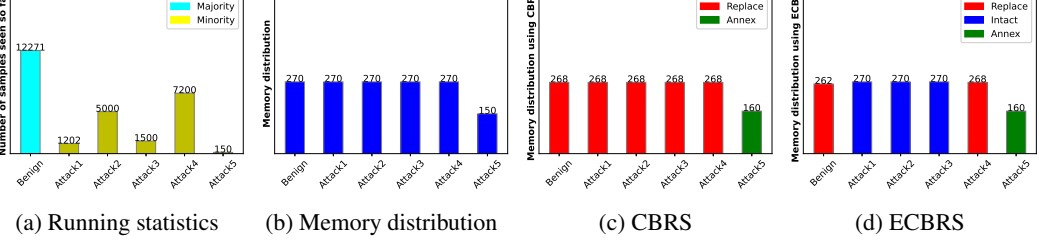

(a) Running statistics    (b) Memory distribution    (c) CBRS    (d) ECBRS

Figure 1: Comparison between CBRS and ECBRS over an imbalanced stream from a CICIDS-2018 dataset setting with a memory size $(\mathcal{M}) = 1500$. (a) Running statistics indicate the number of classwise samples seen so far. (b) Memory distribution represents the strength of each class in buffer memory at a particular instance. Upon the arrival of ten new samples from the class *attack5*, using CBRS, the memory distribution changes to (c). However, using ECBRS, the memory distribution changes to (d). For (c) and (d), the class with the red-colored bar is chosen for replacement, the green-colored bar class receives new samples, and the class with the blue-colored bar remains intact. Detailed illustrations of these configuration changes for CBRS and ECBRS are presented in the supplementary material (SM:A.3).

## 3 Methodology

In this section, we present the proposed ECBRS and PAPA methods. In the ECBRS method, we adopt two definitions introduced in the base paper of CBRS [21]. They are *full* and *largest* classes. When a particular class contains the most instances among all the different classes in the memory,

we call it the *largest*. Two or more classes can be the largest if they have the same size. We call a class *full* if it currently is, or has been in one of the previous time steps, the largest class. Once a class becomes full, it remains so in the future. More details of the CBRS algorithm are given in the supplementary material (SM:A.14).

### 3.1 ECBRS: An Extended Class Balancing Reservoir Sampling Algorithm

The legacy CBRS algorithm assigns higher probability weights to minority class samples using a weighted replay approach. However, these weights are computed based on the number of data samples in the buffer memory (local information). This approach has a limitation when the class imbalance present between the minority classes themselves is severe. This limitation is illustrated in Figure 1, where *running statistics* of each class is the corresponding class running frequency(refer Figure 1a), and *benign* is the maximal (majority) class. Typically, the *benign class* samples must be chosen for replacement whenever new minority class samples arrive. However, CBRS treats all classes equally for a replacement to accommodate newly arrived *attack5* (minority) class samples (instead of the benign class). This is because CBRS relies on local information for sample replacement. From Figure 1b, all class samples appear to be the majority classes in the buffer memory distribution. As a result, different class samples in memory are uniformly selected for sample replacement (refer Figure 1c, in which red-colored classes are chosen for sample replacement for accommodating newly arrived *attack*5 class samples). This choice will have an adverse impact on large-scale training, where the class imbalance between the minority classes is significant. Motivated by this, we extended the CBRS to rely on each class's running statistics (global information) for replacement decision-making. Our method (ECBRS) will always prefer the class with higher global information until a threshold $\gamma$ (refer Equation 1), later it will choose the class with the next highest running statistic value, thereby ensuring that a majority class sample is replaced in the memory buffer. In summary, we use original class labels to organize the buffer memory to learn SBCP and choose class samples in memory with higher running statistic values for the replacement to accommodate newly arriving samples.

Further, the class imbalance among minority classes poses another challenge known as the conflict of equal weights. It will occur whenever the majority (maximal) class samples are completely replaced

---

**Algorithm 1** Memory population for ECBRS

**Input:** data stream: $(x_i, y_i)_{i=1}^n$, number of currently stored instances of class (c≡ $y_i$): $m_c$, number of stream instances of class c ≡ $y_i$ encountered so far: $n_c$

**for** $i = 1$ **to** $n$ **do**
  **if** memory is **not** filled **then**
    store $(x_i, y_i)$;
  **else**
    **if** $y_i$ is **not** a full class **then**
      select a class that is the largest, having higher running statistics value and non-zero samples with $m_c \geq \gamma(c)$ in the buffer. Otherwise, select a class with the next higher running statistic value with $m_c \geq \gamma(c)$;
      overwrite the selected class sample with $(x_i, y_i)$;
    **else**
      sample u ∼ Uniform(0,1);
      **if** u $\leq m_c/n_c$ **then**
        pick a stored instance of class c ≡ $y_i$ at random and replace it with $(x_i, y_i)$;
      **else**
        Ignore $(x_i, y_i)$;
      **end if**
    **end if**
  **end if**
**end for**

---

or underrepresented in the buffer memory. So, undermining the chosen minority class based on global information will lead to the loss of the essential samples in combating the class imbalance problem. This drawback can be mitigated by disallowing the undermining of the minority class beyond a certain threshold $\gamma$. In ECBRS method, we choose this parameter based on global information. Thus, we guarantee that attack class samples can never be replaced beyond this threshold. From Figure 1d, it can be observed that the proposed ECBRS initially chose benign class samples for replacements to accommodate newly arriving *attack*5 class samples, as it has the highest running statistics value. In the process of sample replacement, once it reaches the threshold of the benign class ($\gamma(.)$= 262), ECBRS selects the class with the next highest running static value (*attack*4). Thus remaining, different class samples (colored blue) will remain intact. We compute $\gamma(\cdot)$ (for class $i$) as the expected number of samples to be present based on the global information, as shown in Equation 1.

$$\gamma(i) = m \times w(i), \tag{1}$$

where $i$ is the class index, $w(i) = \frac{\exp(-n_i)}{\sum_j \exp(-n_j)}$, $m$ is the buffer memory capacity, $\gamma(i)$, $n_i$ are the threshold and running class frequency for class $i$, respectively, and $w(i)$ is the weight associated with class $i$. $w(i)$ is the softmax of the negative running

frequencies favoring the minority classes. In our experiments, we also observe that computing a single loss over the replayed and stream samples achieves better performance compared to CBRS (which involves computing two-component loss). One pragmatic reason for this behavior could be that concatenating replay samples with stream samples achieves better batch-wise class balance. We present the pseudo-code of the proposed ECBRS in Algorithm 1.

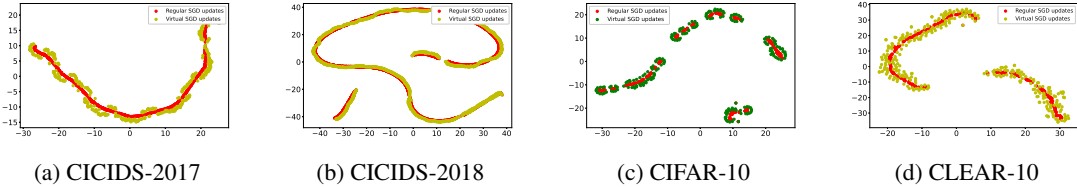

| (a) CICIDS-2017 | (b) CICIDS-2018 | (c) CIFAR-10 | (d) CLEAR-10 |

Figure 2: t-SNE visualization of regular SGD updates and virtual SGD updates. An MLP is trained on the CICIDS-2017 and CICIDS-2018 datasets. A ResNet-18 (pretrain [46]) model is trained on the CIFAR-10 and CLEAR-10 datasets.

## 3.2 Perturbation Assistance for Parameter Approximation (PAPA)

Memory replay-based CL techniques such as MIR and GMED select a subset of samples from the buffer memory based on the loss incurred during virtual stochastic gradient descent (SGD) parameter (VSP) updates. Samples that incur a higher loss are preferred. These updates are computed using the current batch of the incoming stream samples. Further, these updates are ignored after the subset selection. For lengthy data streams (used to train very deep models), these frequent VSP updates result in significant computational overhead (due to SGD operations). Each SGD involves computing the gradients for larger weight matrices, leading to increased computational overhead.

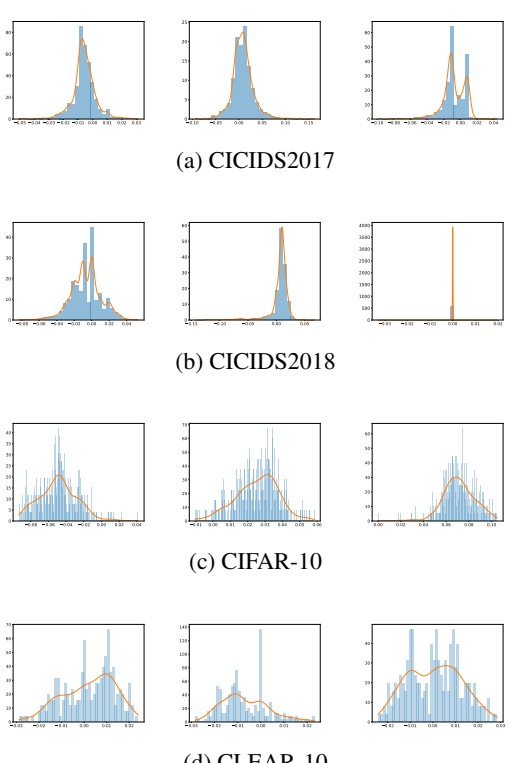

(a) CICIDS2017

(b) CICIDS2018

(c) CIFAR-10

(d) CLEAR-10

Figure 3: For randomly chosen model parameters, the plots show the distribution of the error between the regular SGD parameter updates and virtual SGD parameter updates on various datasets. In (a) and (b), an MLP is trained on CICIDS-2017 and CICIDS-2018. In (c) and (d), a ResNet-18 is trained on CIFAR-10 and CLEAR-10.

Motivated by this problem, we try to understand the relationship between the regular and virtual SGD update operations. Towards this, we plot various t-SNE visualizations on CICIDS-2017, CICIDS-2018, CIFAR-10, and CLEAR-10 datasets. The CIDIDS-2017 and CICIDS-2018 datasets are used to train an MLP, and the CIFAR-10 and CLEAR-10 datasets are used to train a ResNet-18 model (refer Figure 2). We make the following observations from these plots: virtual SGD parameter update is a slowly varying process. In other words, the VSP overlaps with or is scattered closely around the regular SGD updates. One way to capture this notion is to sufficiently *perturb* the regular SGD parameter (RSP) updates to estimate the VSP updates. Then, the next question is: *how to quantify the perturbation required for a given task?*

To quantify the perturbation, we model the error (difference between the VSP and RSP) distribution of randomly chosen model parameters on a diverse set of benchmarks. These are shown in Figure 3. We observe that the error values of the chosen parameters exhibit a skewed Gaussian (heavy-tailed) distribution. We model this error distribution using a two-component Gaussian mixture model (GMM). However, modelling individual error components with a GMM is a time-consuming process. So, we model the joint distribution $\mathcal{P}(\Theta_e)$ of error values of all the parameters using a GMM, whose marginals remain Gaussian. As a result, each parameter-level error distribution is preserved.

Thus, our approach for approximating the VSP is mathematically formulated as a simple additive model which is given in Equation 2.

$$\Theta_{vpu} = \Theta_{rpu} + \mathcal{Z} \tag{2}$$

The perturbation $\mathcal{Z}$ is drawn from $\mathcal{P}(\Theta_e)$ where $\mathcal{P}(\Theta_e) = \pi_1 \mathcal{N}(\Theta_1|\mu_1, \Sigma_1) + \pi_2 \mathcal{N}(\Theta_2|\mu_2, \Sigma_2)$, where $\mu_1, \mu_2$, are the mean vectors and $\Sigma_1, \Sigma_2$ are the covariance matrices of the two Gaussian components, respectively. $\pi_1, \pi_2$ are the mixing coefficients and $\Theta_{rpu}$ is the most recent regular parameter update.

**GMM Training:** We use the MIR algorithm for the first CL task in our proposed approach to estimate the error distribution (ED). This ED is used to train the GMM once, and this GMM is used in all remaining tasks. Our empirical study also confirms that different **first** task (used to construct ED) has no adverse effect on the performance results (refer to **SM:A.10.2**). The pseudo-code of the proposed PAPA method is outlined in Algorithm 2.

## 4    Experiments and Analysis

**Algorithm 2** Perturbation assistance for parameter approximation

1: **Input:** data stream $(x_i, y_i)_{i=1}^n$, initial task $T_1$, maximally interfered retrieval algorithm $\mathcal{MIR}(\cdot)$, gaussian mixture model with two components $\mathcal{GMM}$, buffer memory $\mathcal{M}$, perturbation array $\mathcal{P}$, policy to update buffer memory $\rho$, function approximator $f(x, y : \Theta)$, parameters after updating on previous batch of samples $\Theta_{rpu}$.
2: **for** the task $T_1$ **do**
3:     sample batch $\mathcal{B}_j \sim T_1$
4:     render $\Theta_{j-1}$; parameters after updating on $\mathcal{B}_{j-1}$
5:     Run $\mathcal{MIR}(\mathcal{B}_j)$
6:     store $\Theta_j$; parameters after updating on $\mathcal{B}_j$
7:     store $(\Theta_j - \Theta_{j-1})$ to $\mathcal{P}$
8: **end for**
9: train a $\mathcal{GMM}$ using $\mathcal{P}$
10: **for** remaining tasks **do**
11:     sample batch $\mathcal{B}$ from current task
12:     sample $\mathcal{Z} \sim \mathcal{GMM}$
13:     $\Theta_{vpu} = \Theta_{rpu} + \mathcal{Z}$
14:     $\mathcal{S} \leftarrow$ compute **interfere** samples using $\Theta_{vpu}$ on $\mathcal{M}$
15:     $\mathcal{B}_{int} \leftarrow$ subsample from $\mathcal{S}$ based on certain criteria
16:     $\mathcal{B}_{new} = \mathcal{B} \cup \mathcal{B}_{int}$
17:     train $f(\cdot : \Theta)$ with $\mathcal{B}_{new}$
18:     update $\mathcal{M}$ with $\rho$ using $\mathcal{B}$
19: **end for**

**Datasets and Tasks Formulation:** We conduct the experiments in the domain-incremental (class labels space is fixed to benign and attack, whereas data space horizon may change) learning approach, similar to the prior study [20]. We created tasks by dividing the benign data and combining them with attack class samples. This approach ensures that each task contains a mix of benign and attack data, maintaining the class imbalance resembling real-world network traffic. We created five tasks for KDDCUP'99 [25] and NSL-KDD [47], ten for CICIDS-2017/2018 [22, 48], nine for UNSW-NB [23, 49, 50, 51, 52], and ten for AnoShift [26] benchmark.

Existing network intrusion detection benchmarks are artificially created over short time periods and may not exhibit natural distribution shifts [13] (except AnoShift). Distribution shifts are quantified by the optimal transport dataset distance (OTDD) (refer to Table 2). Large OTDD values indicate a higher distribution shift. More details on *how* OTDD values are computed for each dataset are available in **SM:A.7**. Due to the absence of distribution shifts in NIDS datasets, the proposed approaches were validated using computer vision (CV) CL benchmarks. Furthermore, the ease of visualizing distribution shifts makes CV benchmarks suitable for validating the proposed approaches. For CV benchmarks, we randomly selected one or more classes as the attack classes. We split and distributed the attack class data among the remaining benign classes. This way of class splitting ensures experiments formulated using vision benchmarks are identical to the intrusion detection experiments. Specifically, we created nine tasks for SVHN [27] and CIFAR-10 [28]. Similarly, we created ten tasks for the CIFAR-100 [28] benchmark using the coarse/super class labels. The data spanned over ten years for the CLEAR-10/100 [29] benchmark and contained all the classes' natural temporal evolution for each year. We created ten tasks as the data spans a decade. We also maintained **1:100** class imbalance ratio per task in CIFAR-100, CLEAR-10, and CLEAR-100 experiments. More details about datasets, preprocessing (and feature selection), and task formulations are presented in **SM:A.4, SM:A.5**, and **SM:A.6**.

**Baselines:** We compare the proposed ECBRS method with elastic weight consolidation (EWC) [41], synaptic intelligence (SI) [42] (regularization methods), memory(+gradient)-based algorithms like gradient episodic memory(GEM) [53], A-GEM [54], gradient-based sample selection (GSS-greedy) [55], and rehearsal-based methods like MIR [30], and CBRS [21]. We compare the proposed PAPA with the MIR algorithm in all facets. We deliberately omit partition reservoir sampling [45] and GMED [31]

as the former approach is designed for the multi-label classification setting, and example editing of the latter one is trivial for intrusion data. Additionally, we do not consider INSOMNIA [56], as this work does not operate in the explicit continual learning setup and relies on existing works to identify distribution shifts and pseudo-label generation.

**Architecture details:** The ResNet-18 [46] architecture (pre-trained on Imagenet) is used as the backbone for SVHN, CIFAR-10/100 and CLEAR-10/100, followed by a fully connected network. We use a multi-layer, fully connected network for the NID datasets. We use RMSProp optimizer for intrusion detection and SGD with Nesterov momentum 0.9 and weight decay $10^{-4}$ for computer vision benchmarks, respectively. A batch size of 128 is used for MNIST, SVHN, CIFAR-10/100, 10 for CLEAR-10/100, and 1024 for the rest of the datasets except for NSL-KDD, for which it is 500. More details on batch size, architecture, hyperparameters, and implementation are presented in **SM:A.11**.

Table 2: Average optimal transport dataset distance (OTDD) values of selective intrusion detection and computer vision benchmark datasets.

| Dataset | Avg. OTDD values |
|---|---|
| CICIDS-2018 [18] | 0.0508 |
| CICIDS-2017 [18] | 0.1527 |
| CTU-13 [24] | 4.44 |
| CIFAR-10 [28] | 231 |
| CIFAR-100 [28] | 3139 |
| AnoShift [26] | 7189 |

**Memory size ($\mathcal{M}$):** We use $\mathcal{M}$=13333 for the CICIDS-2017, 2018 and AnoShift datasets. For the KDDCUP'99 dataset, $\mathcal{M}$=5333, and $\mathcal{M}$=1333 for the NSL-KDD dataset. $\mathcal{M}$=500 for the SVHN and CIFAR-10/100 datasets. For the CLEAR-10 dataset, $\mathcal{M}$=666, and $\mathcal{M}$=2666 for CLEAR-100 datasets. 75% of the buffer memory samples are used for replay. Additional details and an ablation study on $\mathcal{M}$ are presented in **SM:A.10.2**.

**Evaluation metrics:** We chose metrics similar to [26, 15]. Specifically, we use the receiver operating characteristic area under the curve (ROC-AUC) metric and the precision-recall area under the curve (PR-AUC) metric for both benign and attack data denoted as PR-AUC (B) and PR-AUC (A), respectively.

Table 3: Performance results comparison of the proposed ECBRS method with the baselines on network intrusion detection and computer vision benchmarks. We report the arithmetic mean with each experiment repeated five times independently. The performance results of the MIR using ECBRS as a memory population method are highlighted in light grey.

| | KDDCUP'99 | | | NSL-KDD | | | CICIDS-2017 | | | CICIDS-2018 | | |
|---|---|---|---|---|---|---|---|---|---|---|---|---|
| Baseline Methods | PR-AUC (A) | PR-AUC (B) | ROC-AUC | PR-AUC (A) | PR-AUC (B) | ROC-AUC | PR-AUC (A) | PR-AUC (B) | ROC-AUC | PR-AUC (A) | PR-AUC (B) | ROC-AUC |
| EWC [41] | 1.000 | 0.694 | 0.995 | 0.949 | 0.852 | 0.927 | 0.617 | 0.766 | 0.608 | 0.740 | 0.762 | 0.505 |
| SI [42] | 1.000 | 0.793 | 0.997 | 0.949 | 0.885 | 0.930 | 0.812 | 0.878 | 0.868 | 0.804 | 0.826 | 0.744 |
| GEM [53] | 1.000 | 0.918 | 0.999 | 0.968 | 0.941 | 0.959 | 0.993 | 0.988 | 0.991 | 0.739 | 0.762 | 0.502 |
| A-GEM [54] | 1.000 | 0.892 | 0.999 | 0.958 | 0.925 | 0.946 | 0.84 | 0.852 | 0.696 | 0.738 | 0.762 | 0.5 |
| GSS-greedy [55] | 1.000 | 0.680 | 0.997 | 0.951 | 0.881 | 0.933 | 0.807 | 0.827 | 0.742 | 0.8215 | 0.762 | 0.664 |
| MIR [30] | 1.000 | 0.767 | 0.996 | 0.923 | 0.861 | 0.918 | 0.785 | 0.840 | 0.798 | 0.737 | 0.762 | 0.5 |
| CBRS [21] | 1.000 | 0.896 | 0.999 | 0.929 | **0.969** | 0.960 | 0.999 | 0.999 | 0.999 | 0.999 | 0.999 | 0.998 |
| **ECBRS (ours)** | **1.000** | **0.982** | **0.999** | **0.970** | 0.967 | **0.969** | **1.00** | **0.999** | **0.999** | **0.999** | **0.999** | **0.998** |
| MIR + ECBRS | 1.000 | 0.929 | 0.999 | 0.958 | 0.966 | 0.965 | 1.00 | 1.00 | 0.999 | 0.994 | 0.994 | 0.992 |

| | UNSWNB-15 | | | CTU-13 | | | AnoShift | | | SVHN | | |
|---|---|---|---|---|---|---|---|---|---|---|---|---|
| Baseline Methods | PR-AUC (A) | PR-AUC (B) | ROC-AUC | PR-AUC (A) | PR-AUC (B) | ROC-AUC | PR-AUC (A) | PR-AUC (B) | ROC-AUC | PR-AUC (A) | PR-AUC (B) | ROC-AUC |
| EWC | 0.925 | 0.823 | 0.913 | 1.00 | 1.00 | 0.999 | 0.543 | 0.566 | 0.550 | 0.971 | 0.953 | 0.964 |
| SI | 0.985 | 0.989 | 0.990 | 1.00 | 1.00 | 0.999 | 0.620 | 0.609 | 0.631 | 0.970 | 0.954 | 0.963 |
| GEM | 0.998 | 0.995 | 0.994 | 1.00 | 1.00 | 0.999 | 0.880 | 0.900 | 0.902 | 0.977 | 0.964 | 0.973 |
| A-GEM | 0.750 | 0.750 | 0.500 | 0.750 | 0.750 | 0.500 | 0.846 | 0.900 | 0.883 | 0.977 | 0.962 | 0.971 |
| GSS-greedy | 0.949 | 0.848 | 0.959 | 1.00 | 1.00 | 0.999 | 0.742 | 0.744 | 0.753 | 0.980 | 0.968 | 0.975 |
| MIR | 0.886 | 0.807 | 0.855 | 0.950 | 0.950 | 0.899 | 0.655 | 0.609 | 0.620 | 0.972 | 0.961 | 0.967 |
| CBRS | **0.999** | **0.999** | **0.999** | 1.00 | 0.999 | 0.999 | 0.949 | 0.939 | 0.941 | 0.954 | 0.957 | 0.955 |
| **ECBRS (ours)** | **0.999** | **0.999** | **0.999** | **1.00** | **1.00** | **0.999** | **0.949** | **0.944** | **0.948** | 0.978 | 0.968 | 0.974 |
| MIR + ECBRS | 0.999 | 0.999 | 0.999 | 1.00 | 1.00 | 0.999 | 0.942 | 0.928 | 0.934 | **0.980** | **0.972** | **0.976** |

| | CIFAR-10 | | | CIFAR-100 | | | CLEAR-10 | | | CLEAR-100 | | |
|---|---|---|---|---|---|---|---|---|---|---|---|---|
| Baseline Methods | PR-AUC (A) | PR-AUC (B) | ROC-AUC | PR-AUC (A) | PR-AUC (B) | ROC-AUC | PR-AUC (A) | PR-AUC (B) | ROC-AUC | PR-AUC (A) | PR-AUC (B) | ROC-AUC |
| EWC | 0.931 | 0.917 | 0.925 | 0.644 | 0.603 | 0.644 | 0.858 | 0.828 | 0.835 | 0.770 | 0.764 | 0.778 |
| SI | 0.929 | 0.915 | 0.925 | 0.645 | 0.614 | 0.635 | 0.847 | 0.833 | 0.835 | 0.766 | 0.753 | 0.772 |
| GEM | 0.931 | 0.919 | 0.928 | 0.653 | 0.626 | 0.643 | 0.858 | 0.852 | 0.848 | 0.818 | 0.800 | 0.813 |
| A-GEM | 0.925 | 0.903 | 0.916 | 0.638 | 0.591 | 0.638 | 0.853 | 0.822 | 0.833 | 0.772 | 0.766 | 0.780 |
| GSS-greedy | 0.944 | 0.932 | 0.940 | 0.659 | 0.646 | 0.662 | 0.881 | 0.850 | 0.861 | 0.733 | 0.691 | 0.721 |
| MIR | 0.941 | 0.936 | 0.939 | 0.640 | 0.640 | 0.636 | 0.890 | 0.885 | 0.887 | 0.837 | 0.800 | 0.82 |
| CBRS | 0.883 | 0.889 | 0.890 | 0.572 | 0.572 | 0.572 | 0.941 | 0.927 | 0.931 | 0.841 | 0.789 | 0.82 |
| **ECBRS (ours)** | **0.953** | **0.941** | **0.948** | 0.663 | 0.611 | 0.663 | 0.937 | 0.926 | 0.926 | **0.854** | **0.807** | **0.831** |
| MIR + ECBRS | 0.949 | 0.936 | 0.944 | **0.663** | **0.659** | **0.663** | **0.953** | **0.933** | **0.942** | 0.839 | 0.793 | 0.817 |

## 4.1 Quantitative analysis

**ECBRS:** We present the performance results of the proposed ECBRS algorithm and its comparison with the baselines on network intrusion detection and computer vision benchmarks in Table 3. We make the following observations. First, we observe that ECBRS either outperforms or is on par with all the baselines on all twelve benchmarks. Second, the performance results of the ECBRS are enhanced over CBRS when both benign and attack data exhibit frequent distribution shifts,

especially on AnoShift, SVHN, CIFAR-10, CIFAR-100, CLEAR-10, and CLEAR-100 benchmarks. Specifically, ECBRS achieves 7% (avg) and 3% (avg) performance gains in attack and benign sample detection in terms of PR-AUC (A) and PR-AUC (B) values relative to the CBRS on aforementioned benchmarks. Third, GEM, AGEM, and GSS-greedy methods strongly compete with our approach. Specifically, for GEM and A-GEM, the reason for this would be due to the usage of the ring buffer memory organization policy [57, 58] that stores task-wise representative samples, which are used to reduce CF of previous tasks [53, 54], thus allowing positive backward transfer by constraining directions of the gradient projections. However, such a buffer organization policy requires knowing task boundaries apriori. Despite their effectiveness, such a policy is inappropriate in an online task-free continual learning setting like streaming intrusion detection in which task boundaries and the number of tasks are unknown. On the contrary, our approach organizes the memory as a pool in which any memory slot can be chosen for replacement. Eventually, we observe that training time for larger datasets is high for gradient-based methods like GEM and GSS-greedy. As a result, we omit the most recent gradient-based memory replay methods like online corset selection (OCS) [59] in our baselines (refer to **SM:A.10.1** for more details). Besides, ECBRS is an efficient algorithm with respect to total training time, and it outperforms CBRS in total training time on five benchmark datasets. More discussions on training times are available in **SM:A.8**.

**ECBRS as a memory population method:** We can also integrate ECBRS as a memory population technique with other memory replay-based continual learning algorithms. Specifically, we demonstrate this for the MIR algorithm and observe an average performance gain of 12% and 14% in terms of PR-AUC (A) and PR-AUC (B) values relative to MIR. We observe that the performance gains of using ECBRS will increase with large-scale benchmarks, especially on AnoShift, CICIDS-2018, UNSW-NB15, and CICIDS-2017 benchmarks. We achieved an average gain of 30% (PR-AUC (A)) and 31% (PR-AUC (B)) on the aforementioned benchmarks. The results of the MIR+ECBRS algorithm (rows) are highlighted in light grey in Table 3. In the ECBRS experiments, we use the random memory population method for the MIR algorithm.

**Threshold parameter ($\gamma$):** As described earlier, we are also interested in learning the distribution shifts of the benign data. Since there are two distribution shifts to adapt to, one may wonder which one is more important. The answer depends on the problem context; intrusion detection requires paying more attention to the distribution shifts in attack traffic, which can be controlled using the hyperparameter $w(\cdot)$. We conduct extensive experiments on intrusion detection and computer vision benchmarks to understand the behavior of setting the default value for the $w(\cdot)$ Equation 1. We found that all the datasets' performance is consistent for the lower $w(\cdot)$ value, so we used a fixed value of 0.1. Eventually, $w(\cdot)$ will be multiplied by the buffer memory size to obtain the expected number of samples per class ($\gamma$) to be present in the buffer memory. Ablation studies on $\gamma(.)$ are presented in **SM:A.10.1**.

Table 4: Performance comparison of the proposed PAPA method with other baselines on intrusion detection and computer vision benchmarks. We report the arithmetic mean of each evaluation metric with each experiment repeated five times independently.

| | MIR | | | PAPA | | | Training time (in sec.) | | |
| Datasets | PR-AUC (A) | PR-AUC(B) | ROC-AUC | PR-AUC (A) | PR-AUC(B) | ROC-AUC | MIR | PAPA | Scalable efficiency |
|---|---|---|---|---|---|---|---|---|---|
| NSL-KDD | 0.964 | 0.971 | 0.970 | 0.961 | 0.968 | 0.969 | 25.4 | 21.4 | 15.7% |
| CICIDS-2017 | 0.999 | 0.999 | 0.999 | 0.999 | 0.999 | 0.999 | 316.0 | 188.8 | 40.2% |
| CTU-13 | 1.000 | 1.000 | 0.999 | 0.999 | 0.999 | 0.999 | 375.5 | 330.8 | 11.9% |
| KDDCUP'99 | 1.000 | 0.929 | 0.999 | 1.000 | 0.920 | 0.999 | 457.0 | 395.0 | 13.5% |
| UNSW-NB15 | 0.999 | 0.999 | 0.999 | 0.999 | 0.999 | 0.999 | 499.4 | 350.6 | 29.7 % |
| AnoShift | 0.944 | 0.926 | 0.934 | 0.947 | 0.927 | 0.934 | 1300.0 | 900.6 | 30.7% |
| CICIDS-2018 | 0.994 | 0.994 | 0.992 | 0.998 | 0.999 | 0.998 | 9040.0 | 5948.0 | 34.2% |
| CIFAR-100 | 0.663 | 0.659 | 0.663 | 0.673 | 0.647 | 0.672 | 115.2 | 76.7 | 33.4% |
| CIFAR-10 | 0.949 | 0.936 | 0.944 | 0.948 | 0.936 | 0.944 | 133.0 | 108.0 | 18.7% |
| CLEAR-10 | 0.953 | 0.933 | 0.942 | 0.943 | 0.927 | 0.932 | 262.9 | 175.4 | 33.2% |
| SVHN | 0.979 | 0.972 | 0.976 | 0.981 | 0.974 | 0.978 | 296.0 | 208.0 | 29.7% |
| CLEAR-100 | 0.839 | 0.793 | 0.817 | 0.845 | 0.793 | 0.823 | 1706.0 | 1209.0 | 29.1% |

**PAPA:** We present the results of the proposed PAPA algorithm and its comparison with the MIR algorithm in Table 4. In this set of experiments, we use ECBRS as a memory population algorithm for both MIR and PAPA algorithms. Results are presented in the non-decreasing order of their respective training times. Based on performance metrics, our evaluation of the benchmarks demonstrates PAPA as a replacement for a method like MIR, which uses numerous virtual SGD updates. From Table 4, it is also interesting to see that our approach's training time efficiency progresses with the dataset's size. Specifically, we note that PAPA achieves nearly 29%, 30%, 33%, 34%, and 40% scalable efficiency

on the CLEAR-100, AnoShift, CIFAR-100, CICIDS-2018 and CICIDS-2017 datasets, respectively. By scalable efficiency, we mean the total training time saved using the PAPA algorithm compared to the MIR method.

Table 5: Performance comparison of the number of regular and virtual SGD operations required for the MIR and proposed PAPA approach on benchmark datasets. Each experiment is repeated five times independently.

| Datasets | MIR | | PAPA | | Savings | |
|---|---|---|---|---|---|---|
| | Vir.SGD ops | Reg. SGD ops | Vir. SGD ops | Reg. SGD ops | Vir. SGD ops | Total SGD ops |
| NSL-KDD | 1140 | 1578 | 210 | 1740 | 81.5% | 28.2% |
| CICIDS-2017 | 9160 | 15784 | 550 | 10881 | 89.5% | 54.17% |
| UNSW-NB15 | 11915 | 14638 | 600 | 12587 | 94.9% | 50.3% |
| CTU-13 | 13235 | 18007 | 120 | 24658 | 99.0% | 20.6% |
| KDDCUP'99 | 19555 | 17525 | 480 | 24450 | 97.5% | 32.7% |
| AnoShift | 48825 | 53187 | 1200 | 58421 | 97.5% | 41.5% |
| CICIDS-2018 | 30590 | 41234 | 1200 | 36605 | 96.1% | 47.3% |
| SVHN | 3850 | 7143 | 360 | 6267 | 90.6% | 39.7% |
| CIFAR-10 | 1935 | 2526 | 180 | 3028 | 90.6% | 28.0% |
| CIFAR-100 | 1700 | 2436 | 135 | 2253 | 92.0% | 42.2% |
| CLEAR-10 | 500 | 602 | 40 | 607 | 92.0% | 41.2% |
| CLEAR-100 | 3250 | 3250 | 320 | 3848 | 90.1% | 35.8% |

**Virtual SGD parameter updates:** Here, we present the core of the proposed PAPA algorithm in terms of the number of virtual stochastic gradient descent parameter updates. In Table 5, we present the results in the non-decreasing order of the size of the datasets and make the following observations. First, the number of virtual SGD operations in MIR increases with the size of the dataset. On the contrary, the number of these operations is reduced with the benchmark size in our proposed method. Second, the number of virtual SGD operations of the PAPA algorithm per benchmark is always fixed. The fixed number of virtual SGD operations in the PAPA is due to using the MIR algorithm only for the first task.

**Ablation studies:** We conduct various experiments as a part of ablation studies to validate the robustness of the proposed approaches. These include the effect of gradient-based sample selection instead of a random selection (refer to line 177) of the ECBRS performance, the effect of batch size and $\gamma$ on ECBRS performance, robustness to different task orders on the PAPA algorithm, the effect of batch size and buffer memory size ($\mathcal{M}$) on the PAPA algorithm. We present and discuss the results in **SM:A.10**. Further, limitations of the proposed methods are discussed in **SM:A.13**.

**Additional experiments:** We also validate the proposed approaches on the standard unsupervised anomaly detection (SMAP [60], SMD [61], and MSL [60]) benchmarks. Our findings on the supervised dataset are equally valid in this new set of experiments (refer to **SM:A.9**).

## 4.2 Impact of task similarity on virtual SGD updates

All the preceding discussions of the PAPA algorithm focus on leveraging the slowly varying parameter update process to approximate virtual SGD updates. Here, we explain the reason for the slowness in the virtual parameter updates using task similarity. Specifically, we use optimal transport dataset distance (OTDD) [62] between two consecutive tasks to quantify the *task similarity*. Also, an error vector (with a dimension equal to the number of parameters in the model) whose entry *'i'* is equal to the difference

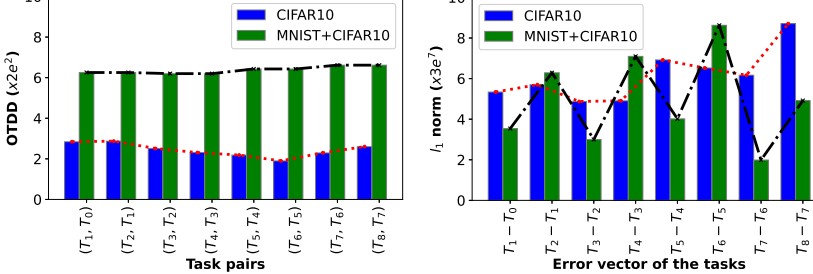

(a) OTD distance between pair of tasks    (b) Error vector values shift visualization

Figure 4: Illustration of the relationship between task similarity and slowness in the parameter update process of the MIR algorithm, using (a) optimal transport dataset distance and (b) visualization of parameter value shift using the $l_1$-norm over the error vector.

in the parameter ($\theta_i$) value prior and after learning a particular task. We perform experiments to compute OTDD and $l_1$-norm of the error vector over the MNIST and CIFAR-10 datasets. We create nine tasks in each of these experiments. We recall that a higher OTDD value signifies lower task similarity. The OTDD value is zero for all the consecutive tasks in the MNIST dataset and therefore ignored in this discussion.

In the second experiment, the tasks are formulated using the CIFAR-10 dataset. The third experiment creates a sequence of two tasks from the MNIST and CIFAR-10 datasets. From Figure 4a, we observe that the OTDD value is higher for the third experiment than the CIFAR-10 experiment. The reason for this behavior is evident since the consecutive tasks in the third experiment learn from two dissimilar datasets. We discover similar behavior in the error vector in which $l_1$-norm over the error vector of the third experiment exhibits a *zig-zag* pattern in Figure 4b. In other words, parameter values go back and forth between two dissimilar tasks. Further, we provided detailed empirical analysis to showcase that the modelling error distribution of the first task using the Gaussian mixture model is sufficient and useful to approximate error distributions of the subsequent tasks. In other words, the occurrence of two dissimilar tasks resulting in different error distributions that can't be modelled by PAPA is rare in the context of NID, as demonstrated in **SM:A.12**. To conclude, assuming overparameterized models, parameter value shifts between the regular SGD and virtual SGD update are low, given the sequence of similar tasks.

## 5    Conclusions and Future Work

In this work, we improved the performance of online memory replay-based CL approaches for network intrusion detection while addressing class imbalance and scalability issues. Specifically, we focused on two popular algorithms viz., CBRS and MIR. The CBRS algorithm is extended (that we call ECBRS) using global information that helps it to maintain accurate class imbalance information. We showed that ECBRS outperforms all the baselines, including CBRS, by maintaining higher minority class samples in the buffer memory. Furthermore, ECBRS is also efficient in terms of the training time incurred as compared to the baselines. We proposed a simple perturbation-assisted parameter approximation (PAPA) method for virtual parameter updation in the MIR algorithm that helps significantly reduce the number of SGD update operations. The efficacy and scalability of these augmentation strategies have been demonstrated on standard network intrusion detection, computer vision, and anomaly detection datasets. In summary, the proposed augmentations to online memory replay-based CL approaches achieved improved performance using fewer computations and thereby at a lower training cost.

As a part of future work, we plan to extend the supervised binary classification setting of the network intrusion detection problem to a supervised multi-class classification problem. Furthermore, we are interested in exploring the semi-supervised techniques for intrusion detection to understand the challenges in the context of class imbalance and distribution shifts in conjunction with open-world learning, explainability, and adversarial robustness settings.

## Acknowledgments and Disclosure of Funding

We extend our appreciation to Aman Panwar and Shreya Kumar, both are undergrads at IIT Hyderabad, for their meaningful contributions during the rebuttal period. Additionally, our gratitude goes out to our former undergrads Akash Tadawai, Reetu Vinta, and Thushara Tippireddy, who played instrumental roles in shaping this work during its initial phases. Finally, we express our sincere thanks to all the anonymous reviewers for their constructive feedback and invaluable suggestions which helped a lot in improving this work. SKA thanks Google for the generous travel grant and the Ministry of Education, Government of India, for the financial support.

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
