# Augmented Memory Replay-based Continual Learning Approaches for Network Intrusion Detection

**Suresh Kumar Amalapuram**[*]**, Sumohana S Channappayya**[#]**, and Bheemarjuna Reddy Tamma**[*]
Department of Computer Science and Engineering[*], Electrical Engineering[#]
Indian Institute of Technology Hyderabad, India
`{cs19resch11001,sumohana,tbr}@iith.ac.in`

## A    Appendix

In this appendix, we present additional details of our proposed work which we could not accommodate in the main paper due to space constraints. Specifically, we shed more light on the following aspects:

- Network intrusion detection system
- Continual learning with shallow methods
- Detailed illustration of configuration changes
- Datasets details
- Data preprocessing and feature selection
- Task formulation
- Task similarity via optimal transport dataset distance
- Training time comparison of the proposed ECBRS with the baselines
- Additional experiments with anomaly detection datasets
- Ablation studies
- Implementation, hardware details, and hyperparameter selection
- Occurrence of task dissimilarity between two different tasks is rare
- Limitations and broader impact

### A.1    Network intrusion detection system

A prototype architecture of network intrusion detection (NID) training and inference system is given in Figure 1. NID comprises two parts: the training module and the anomaly detection engine. The core functionality of the training module is to build the model for intrusion detection using various training datasets. We are building a continual learning network-based intrusion detection model in our work. The training can be periodic or triggered by an event like decay in intrusion detection accuracy. The entire training process can be performed in parallel without affecting the inference process using the MLOps platform for stream processing. After training, the model is deployed to the anomaly/intrusion detection engine. The anomaly detection engine is the visible component of the entire system. It has an in-built feature extractor to extract the essential features from the incoming traffic on the fly. These features are fed to the anomaly detection engine to identify anomaly pattern(s). Further, the proposed model does not require colossal system infrastructure (with a lot of memory and processing resources) as it uses a simple multi-layer perceptron (with about 5 to 6 hidden layers). This MLP architecture has low complexity (capacity) compared to larger models like ResNet with stacked convolution operations. Therefore, our model can also be deployed on edge devices with limited resources. Furthermore, our solution is based on neural networks models, and we recommend using MLOps for low-latency inference in real-world deployments.

37th Conference on Neural Information Processing Systems (NeurIPS 2023).

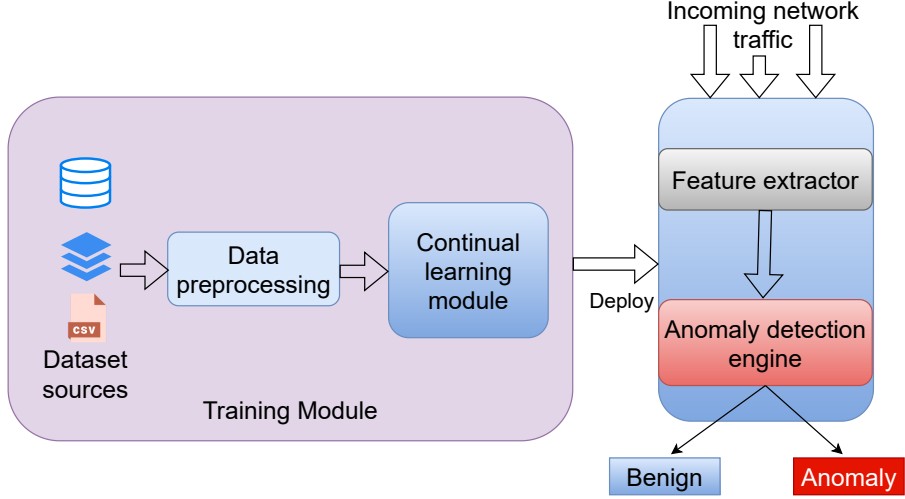

Figure 1: An overview of the continual learning-based network intrusion detection system.

## A.2 Continual learning with shallow methods

In our work, shallow methods are the non-neural network-based approaches. These include methods like random forest, support vector machine, logistic regression, etc. Our empirical observation suggests that using shallow methods in continual learning may not be appropriate for two reasons (i) shallow methods will exhibit higher catastrophic forgetting of the minority classes of the previous tasks and (ii) they still require maintaining samples of previous tasks in buffer memory to handle class imbalance.

We now present the experiments conducted using the random forest in the continual learning setting on CICIDS-2017 and CICIDS-2018 datasets. Specifically, we attempt to understand the implications of using shallow methods on the performance of the attack class (minority) and the overall performance of all previous tasks using the backward transfer (BWT) metric [1]. First, we define the BWT metric and show how it is computed. BWT is the influence that learning a task 't' has on the performance of a previous task $k < t$, and it can be either positive or negative. Positive BWT occurs when acquiring knowledge in a particular task enhances one's performance in a prior task. Conversely, negative BWT occurs when learning a task diminishes proficiency in prior tasks. Notably, significant negative backward transfer is often termed as **catastrophic forgetting** (CF). To compute BWT, we assume implicit access to the test set for each task 't.' After the model completes learning the task 't', we evaluate the test performance on all the 't' tasks and create a matrix $\mathcal{R} \in \mathbf{R}^{t \times t}$. Each entry $\mathcal{R}[i, j]$ is the test performance measure of the task 'j' after learning until the task 'i.' The BWT for a particular task 'p' is computed as follows:

$$\text{BWT}[p] = \frac{1}{p-1} \sum_{j=1}^{p-1} \mathcal{R}[p, j] - \mathcal{R}[j, j]$$

**Performance on minority class (attack class)**: Here, we study the impact of using the random forest method in the CL setting on the performance of minority classes using the BTW metric. Specifically, the test performance is evaluated using the PR-AUC metric after the corresponding $\mathcal{R}$ matrix is constructed. Using the $\mathcal{R}$ matrix we computed the BWT values as presented in Table 1 on CICIDS-2017 and CICIDS-2018 datasets. We make the following observations: first, most of the BWT values are negative, indicative of higher CF. Second, while augmenting the random forest training process with additional buffer memory reduced negative BWT, it still exhibited significant CF. To conclude, random forest-based CL training suffers from higher CF even with augmented memory. As a result, shallow methods like random forest may not be appropriate under a continual learning paradigm.

**Overall performance:** Here, we study the impact of using the random forest method in the CL setting on the overall performance of all tasks using the BTW metric. Specifically, the test performance

Table 1: Backward negative transfer (catastrophic forgetting) of the minority (attack) classes using the random forest method in the continual learning setting. This set of experiments uses the PR-AUC metric to compute the test performance of all tasks.

| Dataset | Buffer memory size | Task1 | Task2 | Task3 | Task4 | Task5 | Task6 | Task7 | Task8 | Task9 | Task10 | Task11 |
|---|---|---|---|---|---|---|---|---|---|---|---|---|
| CICIDS-2017 | - | 0. | -0.98 | -0.72 | -0.54 | -0.35 | -0.45 | -0.38 | -0.37 | -0.38 | -0.39 | -0.39 |
| | 11000 | 0. | -0. | -0.07 | -0.11 | -0.08 | -0.25 | -0.13 | -0.1 | -0.21 | -0.19 | -0.24 |
| | 110000 | 0. | -0. | -0.07 | -0.09 | -0.07 | -0.24 | -0.11 | -0.09 | -0.21 | -0.23 | -0.19 |
| CICIDS-2018 | - | 0. | -0.45 | -0.9 | -0.41 | -0.42 | -0.4 | -0.71 | -0.66 | -0.45 | -0.88 | - |
| | 10000 | 0. | 0. | -0.16 | -0.24 | -0.18 | -0.28 | -0.24 | -0.2 | -0.3 | -0.42 | - |
| | 100000 | 0. | -0. | -0.16 | -0.24 | -0.18 | -0.25 | -0.22 | -0.19 | -0.26 | -0.32 | - |

is evaluated using the F1-score metric, and the corresponding $\mathcal{R}$ matrix is constructed. Using the $\mathcal{R}$, we computed the BWT, and values are presented in Table 2 on CICIDS-2017 and CICIDS-2018 datasets. Previously described drawbacks (in the performance of minority class experiment) are also equally valid in this experiment. To conclude, overall performance suffers from higher CF even with augmented memory. As a result, shallow methods like random forest may not be appropriate under a continual learning paradigm.

Table 2: Backward negative transfer (catastrophic forgetting) of the overall performance using the random forest method in the continual learning setting. This set of experiments uses the F1-score metric to compute the test performance of all tasks.

| Dataset | Buffer memory size | Task1 | Task2 | Task3 | Task4 | Task5 | Task6 | Task7 | Task8 | Task9 | Task10 | Task11 |
|---|---|---|---|---|---|---|---|---|---|---|---|---|
| CICIDS-2017 | - | 0. | -1. | -0.99 | -0.88 | -0.99 | -0.9 | -0.97 | -0.96 | -0.98 | -0.99 | -0.95 |
| | 11000 | 0. | -0. | -0.09 | -0.27 | -0.19 | -0.35 | -0.28 | -0.2 | -0.33 | -0.26 | -0.42 |
| | 110000 | 0. | -0. | -0.09 | -0.19 | -0.13 | -0.3 | -0.22 | -0.17 | -0.31 | -0.25 | -0.3 |
| CICIDS-2018 | - | 0. | -0.95 | -0.97 | -0.98 | -0.95 | -0.87 | -0.99 | -0.99 | -0.98 | -0.98 | - |
| | 10000 | 0. | -0. | -0.38 | -0.59 | -0.44 | -0.6 | -0.5 | -0.43 | -0.48 | -0.53 | - |
| | 100000 | 0. | -0. | -0.38 | -0.59 | -0.44 | -0.56 | -0.46 | -0.4 | -0.45 | -0.47 | - |

### A.3 Detailed illustration of memory configuration changes

**CBRS**: It always selects the largest class sample for replacement whenever a new non-full/largest class sample arrives. In our study, *attack5* class is the minority/non-full class, and the remaining classes (*benign, attack1, attack2, attack3,* and *attack4*) are full classes. CBRS chooses one of the full class samples for replacement whenever a new *attack5* class sample arrives. For instance, the *attack4* class sample is chosen for the first arriving sample (illustrated in Figure 2a). As a result, *attack4* class strength in buffer memory will be reduced to 269. Now, on the newly arriving *attack5* sample, CBRS will choose classes other than *attack4*, i.e., *benign, attack1, attack2,* and *attack3* for replacement, as they have higher class strength in the buffer memory, which is 270. Consequently, *benign* class is chosen for replacement (illustrated in Figure 2b). This process continues until all the new arriving *attack5* class samples are accommodated in the buffer memory (illustrated from Figure 2c to Figure 2j). As a result, each of the largest classes in the buffer will be chosen at least once for replacement.

**ECBRS**: It always selects a class sample with higher *running statistics* and class strength in buffer memory greater than $\gamma(\cdot)$ for replacement whenever a new non-full/largest class samples arrive. In our study, *attack5* class is the minority/non-full class, and the remaining classes (benign, attack1, attack2, attack3, and attack4) are full classes. ECBRS will choose *benign* class sample for replacement whenever a new attack5 class sample arrives (illustrated in Figure 3a). As a result, benign class strength in buffer memory will be reduced to 269. Now, on the newly arriving attack5 sample, ECBRS still chose *benign* class for replacement. The benign class sample selection process continues until its class strength reaches the threshold ($\gamma(benign) = 262$) (illustrated from Figure 3b to Figure 3i). Eventually, benign class strength reaches its threshold whenever a tenth new sample from the attack5 class arrives. As a result, ECBRS will choose the class *attack4* for replacement, as it contains the next highest running statistic value, and the remaining classes (attack1, attack2, attack3) are left undisturbed (illustrated in Figure 3j

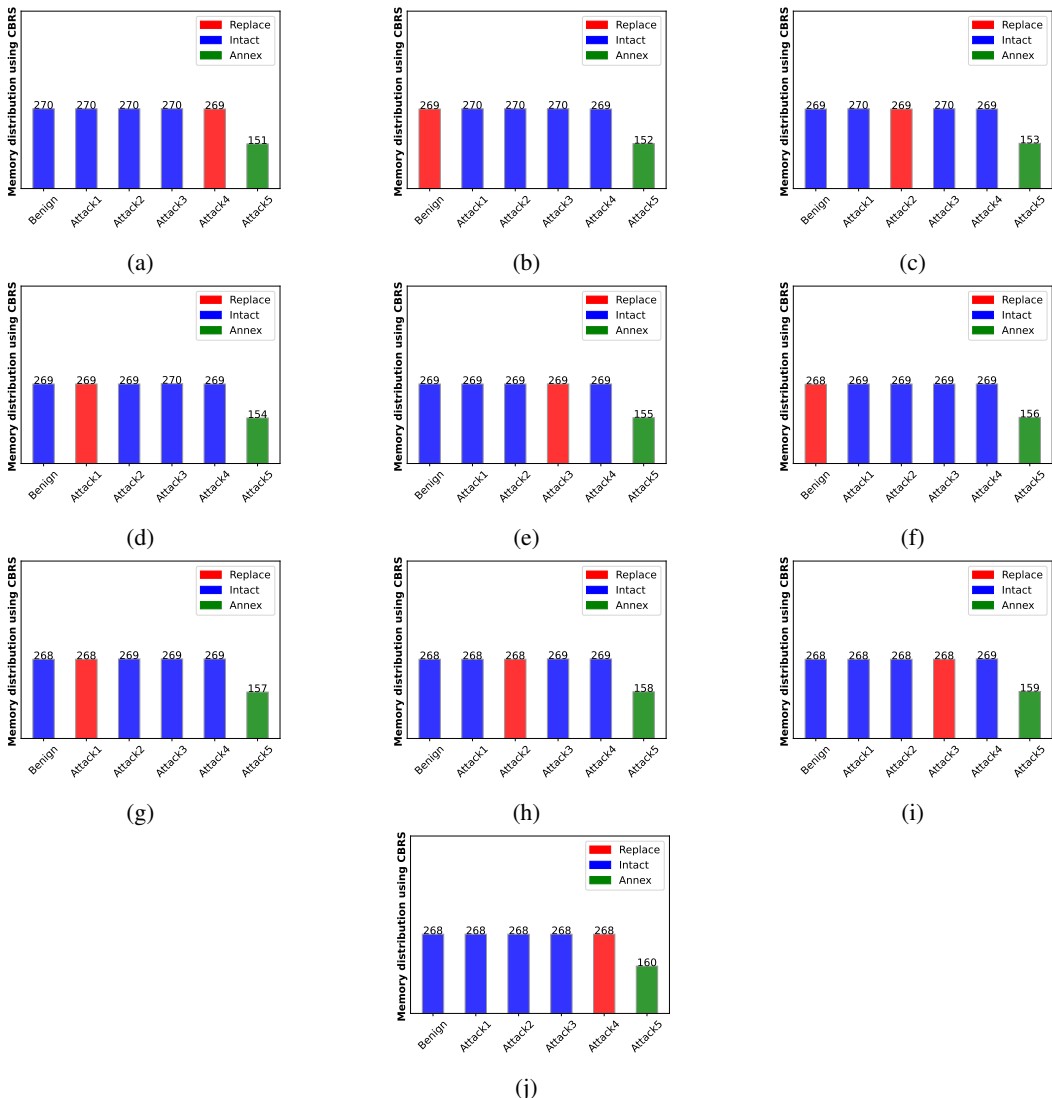

Figure 2: Illustration of variations in the CBRS buffer memory configuration on the arrival of each new sample (of attack4 class). A total of 10 new samples arrived for *attack5* class, and a corresponding memory configuration was shown from (a) - (j). The class with blue-colored bars remains intact, the green-colored bar class received a new sample, and the red-colored bar class is chosen for replacement.

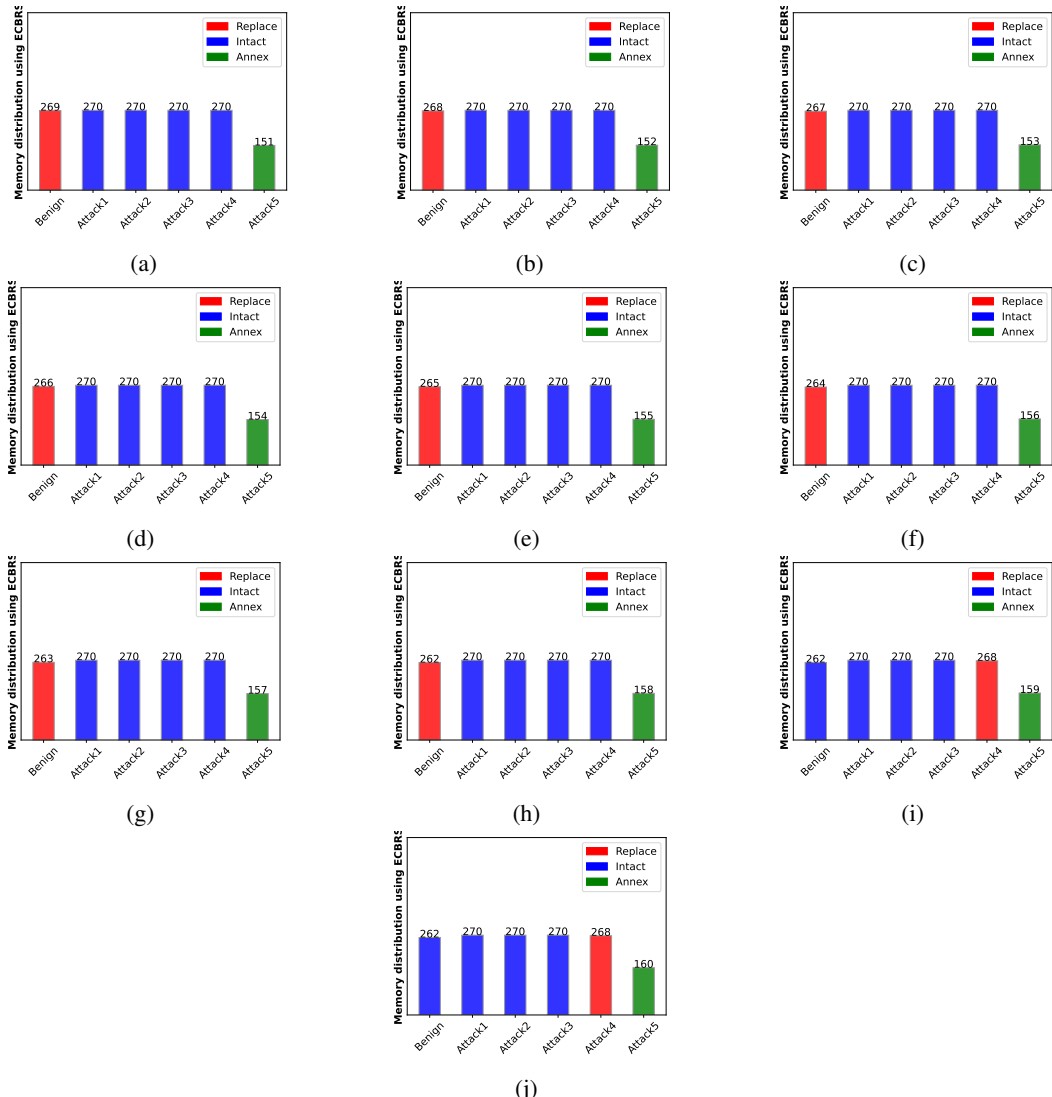

Figure 3: Illustration of variations in the ECBRS buffer memory configuration on the arrival of each new sample (of attack4 class). A total of 10 new samples arrived, and a corresponding memory configuration was shown from (a) - (j). The class with blue-colored bars remains intact, the green-colored bar class received a new sample, and the red-colored bar class is chosen for replacement.

## A.4 Datasets

**KDDCUP'99** [2] is a widely used dataset for network anomaly detection, built using the data captured in the DARPA'98 IDS evaluation program. Its raw data contains seven weeks for captured network traffic used for training and two weeks for data for the test set. Different types of attacks present in this dataset fall under one of the four attack categories (denial of service, user to root, remote to local, and probing). This dataset set has nearly 4.9 million samples, with 41 features each. The train set contains nearly 24 types of attacks, and the test set include additional 14 new different types. Despite these merits, it received a lot of criticism due to redundant records, and the difficulty of learning is low. A new version was created, eliminating the redundancy, called **NSL-KDD**. However, this also still suffers some of the problems raised [3]; this may not be representative of real-world network traffic.

**CICIDS-2017** and **CICIDS-2018** [4] are well-known multiclass up-to-date datasets, curated by the Canadian Institute for Cybersecurity (CIC) [5, 6]. However, the published version released by CIC received a lot of criticism recently due to errors in curating the dataset. The published version of CICIDS2018 only contains 25% of the total flows, mislabelling issues for attack classes, and the presence of class overlap when flow identifiers are removed. In our experiments, we used the recent dataset [4]. The experiments to simulate these datasets are spread over 5 and 10 days, respectively. These datasets contain nearly $2\ million$ and $63\ million$ in training samples, respectively. The CICIDS-2017 is curated using two subnetworks, attack and victim networks, whereas CICIDS-2018 is curated using six subnetworks on the $AWS$ platform [5].

**UNSW-NB15** [7, 8, 9, 10, 11] is another multiclass cybersecurity dataset which contains nine attack classes and one benign class. The simulation period was 16 hours on Jan 22, 2015, and 15 hours on Feb 17, 2015. The testbed architecture used for curating this dataset contains two subnetworks. This dataset has approximately 2.5 million samples, out of which 87% belong to the benign class.UNSW-NB15 dataset originally had nine different classes. However, to enlarge the class imbalance, we split some attack classes to give 14 classes.

**CTU-13** [12] comprises botnet traffic collected at CTU University, Czech Republic, in 2011. It aimed to amalgamate authentic botnet traffic with regular and background traffic for a comprehensive dataset. It consists of thirteen distinct botnet sample captures (scenarios), each involving the execution of specific malware utilizing various protocols and actions. The CTU-13 dataset stands out due to each scenario's manual analysis and labeling. The different botnet scenarios include Menti, Murlo, Neris, Rbot, Virut, NSIS.ay, and Sogou.

**AnoShift** [13] is a new unsupervised anomaly detection benchmark build over kyoto2006+ (intrusion detection dataset). This dataset is collected using various honeypots deployed across five different subnetworks in Kyoto University. All the normal traffic is generated using a mailing server and a DNS server. The malicious network traffic is additionally categorized using three software solutions: a Network-Level Intrusion Detection System, an Antivirus product, and a detector for shellcodes and exploits. We inform the readers that the testbed architecture used to curate this dataset differs from the remaining datasets. However, we are interested in this dataset because it is the most suitable dataset to validate proposed continual learning methods due to its natural temporal variations and time period of ten years. This dataset contains nearly 90% anomalous traffic. It has nearly 806 million records; however, in our work, we restricted it to a subset of this whole benchmark.

**MNIST** [14] is a dataset of handwritten digits ranging from 0 to 9. It has 60,000 training, a test set of 10,000 examples, and each sample has a size of $28 \times 28$.

**SVHN** [15] is a street view housing number dataset. It is a real-world dataset obtained from house numbers in Google Street view images. It has 10 class digits labeled from 0 to 9, each with the size of $3 \times 32 \times 32$.

**CIFAR-10/100** [16] is a labeled dataset derived from the tiny IMAGENET dataset. It has 60,000 colored $32 \times 32$ size training images with 10/100 different classes and 50,000 test images. The classes for the CIFAR-10 dataset are airplane, automobile, bird, cat, deer, dog, frog, horse, ship, and truck. The CIFAR-100 dataset also has twenty $super/coarse$ class labels.

**CLEAR-10/100** [17] is the first continual image classification benchmark with natural temporal evolution in the visual concepts of real-world imagery spanning over ten years. It was built from the existing scale image collection YFCC100M [18]. Authors selected temporally dynamic visual

concepts in the trends and fashion, consumer products, and social events. Further, ten dynamic visual concepts were chosen from the above super category; they are baseball, bus, camera, cosplay, dress, hockey, laptop, racing, sweater, and soccer for the CLEAR-10 dataset. Specifically, the dataset is organized in a yearly granularity containing ten directories (one per year), and each directory contains ten classes for CLEAR-10 and a hundred classes for CLEAR-100 dataset.

### A.5 Data pre-processing & feature selection

This section describes the data processing and feature selection methods used on the benchmark datasets before training.

**KDDCUP'99 and NSL-KDD:** For these datasets, $service$, $is\_host\_login$, $num\_outbound\_cmds$ features were removed as they were found to contain redundant values. After preprocessing, we used 3908372 samples as a training set and 990059 samples for the test for KDDCUP'99. For NSL-KDD, the train and test contain 112752 and 35767 samples, respectively.

**CICIDS2017 and CICIDS2018**: These datasets initially contained over 90 features. After eliminating flow-specific identifiers such as flow ID, source IP, destination IP, source port, destination port, and timestamp, we applied feature engineering using the Pearson correlation coefficient with a threshold of 90%. This process led to retaining around 51 features. With these selected features, min-max normalization is performed before starting the training process. The training and test set sizes of CICIDS-2017 dataset are 1828315 and 271656, whereas for CICIDS-2018, these are 61178191 and 2016897, respectively.

**USNW-NB15**: For this dataset, we removed $srcip$, $sport$, $dstip$, $dsport$, $Stime$, $Ltime$ features. We performed a categorical data encoding for the features like $proto$, $state$, $service$, followed by min-max normalization, after which the training process was invoked with the remaining features. The training and test sizes are 2379387 and 160660, respectively.

**AnoShift-subset**: We did not perform any feature engineering on this dataset, as the preprocessed dataset is readily available with 18 features. The training and test sizes are 9759548 and 1525000, respectively.

**CTU-13**: We used the dataset from [19]. We performed min-max normalization on this dataset before training.

**CIFAR-10 and CIFAR-100**: These datasets are normalized across all three channels using the mean (0.5071, 0.4865, 0.4409) and standard deviation values (0.2673, 0.2564, 0.2762). Following, we applied torchvision transform operations like resize and random crop with a value of 32.

**CLEAR-10 and CLEAR-100**: These datasets are normalized across all three channels using the mean (0.485, 0.456, 0.406) and standard deviation values (0.229, 0.224, 0.225). Following, we applied torchvision transform operations like resize and random crop with a value of 224.

### A.6 Task formulation

In this section, the intuitions behind how a task is created in the *continual learning* context for each benchmark dataset is described. Most CL literature is devoted to the computer vision domain, so creating tasks using vision datasets is well-established in the literature. However, the situation is different for network intrusion detection datasets due to the absence of established procedures for creating tasks for NIDS datasets. Based on our prior experience in the NIDS domain, we recommend following the desiderata: (i) each created task has to contain a large number of benign samples compared to attack samples, (ii) when compared, each task has to exhibit quantifiable distribution shifts and contain longer task sequences; however, most publicly available NIDS datasets contain minimal to no distribution shifts. One possible way to solve this problem is by posing task creation as an optimization problem to maximize the average distribution shifts between tasks, with an additional constraint that each task at least has one attack class and one benign class. Given the complexity of solving this optimization problem, we used a simple heuristic approach to create the tasks that satisfy the above desiderata. In the following, we describe the number of tasks created for each dataset.

**KDDCUP'99** and **NSL-KDD**: We create ten tasks for KDDCUP'99 and five for NSL-KDD.

**CICIDS-2017** and **CICIDS-2018**: We create eleven tasks for CICIDS-2017 and ten for CICIDS-2018.

**CTU-13**: This dataset [19] considers five attack classes (Menti, Murlo, Neris, Rbot, and Virut) samples. We created each task by distributing each attack class sample per task. As a result, a total of five tasks are created.

**UNSW-NB15**: We created a total of nine tasks for this dataset

**AnoShift**: This dataset spread over ten years. We created ten tasks, considering each year as a unit in creating each task. Specifically, the second desideratum for this dataset is naturally obeyed (as already shown in the literature about the natural distribution shifts of this dataset).

**SVHN** and **CIFAR-10**: We randomly select one class as an attack class and all remaining classes as benign. Specifically, for SVHN, $class$ with label 9 is chosen as an attack class, and nine tasks are created, accommodating each of the remaining nine classes(benign) per task. The chosen attack class is randomly split into nine parts and distributed across all the tasks. Similarly, nine tasks are created for the CIFAR-10 dataset with the $truck$ class as an attack class.

**CIFAR-100**: We consider $super$class labels (a total of 20 labels) of CIFAR-100 dataset in our experiments. A single large $benign$ class file is created, combining samples from the class with labels from 0 to 13, and a single $attack$ class file is created with the remaining class labels. After shuffling, the large benign file is split into $ten$ equal parts, and the attack file is split into $ten$ parts by maintaining a $classimbalance$ (CI) ratio of $1 : 100$ between each benign and attack subpart. Thus, when a task is created, combining a single benign and attack class, a CI ratio of 1:100 is maintained per task. Another merit of such a task creation is that, when compared, a distribution shift exists between the benign and attack classes of different tasks. We have created a total of ten tasks for this dataset.

**CLEAR-10**: Similar to CIFAR-100 tasks creation, we create a large $benign$ file considering six classes of data from each year. Specifically, six classes of data from all ten years (of training data) are combined to create the benign file, and a single attack class is created with the remaining four classes of data from all ten years. The large benign class is divided into ten equal subparts, and the attack class is divided into sixteen parts, each containing the number of samples that is 1/100th the number of samples in the benign subpart file to maintain a CI ratio of 1:100 per task. We created a total of ten tasks for this dataset.

**CLEAR-100**: Similar to CLEAR-10 tasks creation, we create a large $benign$ file considering eighty classes of data from each year. Specifically, eighty classes of data from all ten years (of training data) are combined to create the benign file, and a single attack class is created with the remaining twenty classes of data from all ten years. The large benign class is divided into ten equal subparts, and the attack class is divided into sixteen parts, each containing the number of samples that is 1/100th the number of samples in the benign subpart file to maintain a CI ratio of 1:100 per task. We created a total of ten tasks for this dataset.

## A.7 Task similarity via optimal transport dataset distance

In this section, we describe the optimal transport dataset distance (OTDD) [20] for task similarity. OTDD is a geometric distance that compares the similarity between two continual learning tasks. The reason for using the OTDD metric despite other distance metrics lies in its uniqueness as follows: OTDD is model agnostic and it does not involve training.

Now, we describe how to leverage OTDD values to compute the distance between the adjacent tasks (of CL training). We use these computed OTDD values to verify the presence of distribution shifts that happened between the tasks. Eventually, we use the single number per dataset computed as the average of these OTDD values to compare different datasets to quantify the amount of distribution shifts present at a dataset level.

We made the following observation from Table 3. First, datasets like CICIDS-2017, CICIDS-2018, and CTU-13 contain minimal to no distribution shift (DS) in their datasets; on the contrary, the AnoShift dataset contains higher DS compared to CIFAR-10 and CIFAR-100. Second, due to high dimensionality and different types of images presented, the CIFAR-10/100 exhibits DS. Interestingly, unlike other NIDS datasets, AnoShift exhibits higher DS. The reason for this behavior can be bi-

Table 3: Optimal transport dataset distance values computed between adjacent tasks. The average OTDD value is computed as the mean value over all the OTDD values.

| Dataset | (1,2) | (2,3) | (3,4) | (4,5) | (5,6) | (6,7) | (7,8) | (8,9) | (9,10) | Avg. OTDD values |
|---|---|---|---|---|---|---|---|---|---|---|
| CICIDS-2017 | 0.2564 | 0.1821 | 0.2626 | 0.0154 | 0.3031 | 0.3094 | 0.0339 | 0.0078 | 0.0039 | **0.1527** |
| CICIDS-2018 | 0.1760 | 0.1576 | 0.0093 | 0.0052 | 0.0051 | 0.0043 | 0.0042 | 0.0357 | 0.0732 | **0.0523** |
| CTU-13 | 0.9451 | 3.0247 | 9.3560 | - | - | - | - | - | - | **4.4419** |
| AnoShift | 21190.6797 | 4506.2402 | 4617.2363 | 6344.8398 | 9892.0820 | 5103.3872 | 3602.3271 | 2258.5381 | - | **7189** |
| CIFAR-10 | 268.7531 | 265.6069 | 242.938 | 221.597 | 212.276 | 181.975 | 216.785 | 241.152 | - | **231** |
| CIFAR-100 | 1687.2399 | 1686.1848 | 1693.7480 | 1723.9976 | 1698.5425 | 1715.5204 | 1695.5215 | 1686.7532 | 1676.0833 | **1695.95457** |

fold: (i) time duration and (ii) network topology used to curate the dataset. The CICIDS-2017 and CICIDS-2018 datasets are synthetically created over a short period (within days) [21], and the topology contains two subnets (attack and victim networks). However, the AnoShift dataset is curated from the existing Kyoto2006+ dataset that spans over a decade, and the data is logged from 348 honeypots deployed over five different subnets at Kyoto University [21]. Although AnoShift is curated differently from other NIDS datasets, the presence of DS over a long period makes it an ideal dataset to validate the proposed methods in this work.

## A.8  Training time comparisons of the ECBRS with the baselines

Here, we compare the training time required for all the baselines across network intrusion detection and computer vision benchmarking datasets with the proposed ECBRS method, and the results are presented in Table 4. We make the following observations.

1. The training time for gradient-based methods GEM, AGEM, and GSS-greedy increases with the size of the dataset, as they require a lot of gradient computations.

2. CBRS and ECBRS take the least training time on nine datasets. However, ECBRS' training time is lower when compared to CBRS on seven benchmarks out of twelve.

3. ECBRS is efficient regarding training time compared to the baselines on six benchmarking datasets, including large-size datasets like CICIDS-2018 and AnoShift.

4. Empirically, besides CBRS and ECBRS, MIR is the most training time-efficient method among all the baselines.

Table 4: Total training time comparison between the baselines and ECBRS on benchmark datasets. Each experiment is repeated five times independently, and mean values are reported. Training time is reported in seconds.

| Method | KDDCUP'99 | NSL-KDD | CICIDS2017 | CICIDS2018 | UNSW-NB15 | CTU-13 | AnoShift | SVHN | CIFAR-10 | CIFAR-100 | CLEAR10 | CLEAR-100 |
|---|---|---|---|---|---|---|---|---|---|---|---|---|
| EWC | 1089 | 37 | 518 | 18078 | 789 | 674 | 2734 | 366 | 142 | 1197 | 421 | 1376 |
| SI | 948 | 37 | 398 | 16780 | 667 | 658 | 2119 | 460 | 238 | 283 | **204** | **1221** |
| GEM | 1100 | 37 | 489 | 17296 | 801 | 605 | 2716 | 1205 | 638 | 772 | 965 | 6608 |
| A-GEM | 847 | 31 | 407 | 13658 | 669 | 554 | 2022 | 460 | 97 | 558 | 645 | 1337 |
| GSS-greedy | 35445 | 257 | 9212 | 92055 | 6966 | 505 | 36388 | 6111 | 2346 | 4837 | 1456 | 8393 |
| MIR | 420 | 23 | 254 | 7620 | **428** | 398 | 1210 | 217 | 118 | 96 | 265 | 1490 |
| CBRS | **378** | 19 | **190** | 6534 | 478 | **295** | 976 | 159 | 82 | **72** | 613 | 1552 |
| ECBRS | 387 | **16** | 216.6 | **6111** | 452 | 258 | **846** | **112** | **61** | 74 | 534 | 1570 |

## A.9  Additional experiments with anomaly detection datasets

We took the test sets (as the labels are available) of datasets (i.e., SMAP, MSL, and SMD) to validate our proposed approaches. We found that our findings on the supervised dataset are equally valid on the new set of experiments conducted on the test sets of the unsupervised anomaly detection datasets. Now, we will explain in detail the experiment formulation and discuss the results.

**Experiments formulation**: We divide the SMAP and MSL data into six tasks, each containing a mix of anomalous and normal data. Similarly, three tasks were created for the SMD dataset, as there are three groups of entities. Hyperparameter details of the experiments are presented in Table 5. The intuition behind the memory size is to store nearly 1% of the total training sample in the buffer memory, and 75% of the buffer memory samples will be used for replay.

In this new set of experiments, we partially validated the proposed approach on unsupervised datasets with the most suitable baselines. First, we present the performance results of the proposed ECBRS

Table 5: Hyperparameters details of the experiments conducted on unsupervised anomaly detection benchmark datasets.

| Dataset | #tasks | Memory/Replay size | $w(\cdot)$ |
|---|---|---|---|
| SMAP- Soil Moisture Active Passive | 6 | 4500/3375 | 0.1 |
| MSL-Mars Science Laboratory Rover | 6 | 750/560 | 0.1 |
| SMD-Server Machine Dataset | 3 | 7000/5250 | 0.1 |

algorithm in Table 6. ECBRS exhibits a similar performance trend to other datasets. Additionally, when compared to MIR with random replay sample replacement, ECBRS shows superior performance. Moreover, when used as a memory population technique in conjunction with MIR, ECBRS outperforms MIR with the random replay technique.

Table 6: Performance results comparison of the proposed ECBRS method with the baselines on anomaly detection benchmarks. We report the arithmetic mean values with each experiment repeated five times independently. The performance result of the MIR using ECBRS as a memory population method is highlighted in light grey color.

| | SMAP | | | SMD | | | MSL | | |
|---|---|---|---|---|---|---|---|---|---|
| Baseline Methods | PR-AUC (A) | PR-AUC (B) | ROC-AUC | PR-AUC (A) | PR-AUC (B) | ROC-AUC | PR-AUC (A) | PR-AUC (B) | |
| MIR [22] | 0.560 | 0.916 | 0.598 | 0.751 | 0.637 | 0.709 | 0.587 | 0.569 | 0.597 |
| CBRS [23] | 0.713 | 0.717 | 0.738 | 0.902 | 0.932 | 0.922 | 0.718 | 0.645 | 0.683 |
| **ECBRS (ours)** | 0.736 | 0.721 | 0.752 | 0.895 | 0.925 | 0.905 | 0.723 | 0.647 | 0.686 |
| **MIR + ECBRS** | 0.648 | 0.647 | 0.672 | 0.839 | 0.818 | 0.824 | 0.616 | 0.660 | 0.695 |

In another set of experiments, we compare the effectiveness of the proposed PAPA algorithm on unsupervised datasets. The performance results are presented in Table 7 and Table 8. Even here, we note that the performance results trend of the proposed PAPA algorithm on suggested datasets is similar to the experimental results discussed in the main paper. The proposed PAPA approach outperforms MIR on all three datasets. It achieves a maximum of 85% savings in the number of virtual SGD updates, leading to 50% savings in the total number of SGD updates.

Table 7: Performance comparison of the proposed PAPA method with other baselines on anomaly detection benchmarks. We report the arithmetic mean of each evaluation metric with each experiment repeated five times independently.

| | MIR | | | PAPA | | | Training time (in sec.) | | |
|---|---|---|---|---|---|---|---|---|---|
| Datasets | PR-AUC (A) | PR-AUC (B) | ROC-AUC | PR-AUC (A) | PR-AUC (B) | ROC-AUC | MIR | PAPA | Scalable efficiency |
| SMAP | 0.648 | 0.647 | 0.672 | 0.717 | 0.724 | 0.744 | 186.6 | 126.4 | 32.2% |
| SMD | 0.839 | 0.818 | 0.824 | 0.876 | 0.906 | 0.886 | 301.6 | 208.6 | 30.8% |
| MSL | 0.616 | 0.576 | 0.600 | 0.723 | 0.660 | 0.695 | 37.6 | 28.2 | 25.0% |

Table 8: Performance comparison of the number of regular and virtual SGD operations required for the MIR and proposed PAPA approach on benchmark datasets. Each experiment is repeated five times independently.

| | MIR | | PAPA | | savings | |
|---|---|---|---|---|---|---|
| Datasets | Vir.SGD ops | Reg. SGD ops | Vir. SGD ops | Reg. SGD ops | Vir. SGD ops | Total SGD ops |
| SMAP | 16010 | 29361 | 2295 | 20315 | 85.6% | 50.1% |
| SMD | 27740 | 46612 | 5040 | 33105 | 81.8% | 48.6% |
| MSL | 3120 | 4823 | 450 | 3916 | 85.5% | 45.0% |

## A.10  Ablation studies

In this section, we conduct various ablation studies on the hyperparameters of the proposed ECBRS and PAPA methods.

### A.10.1 ECBRS

In this section, we study the impact of varying each of the hyperparameters on the performance of the proposed ECBRS method. Specifically, ablation studies are conducted on random sample selection, threshold parameter $\gamma(\cdot)$, and batch size.

**Gradient based sample selection**: In the proposed ECBRS approach, whenever a new sample belonging to the *full class* arrives, a *randomly* chosen sample from the same class is replaced with a new sample. This experiment replaces random sample selection with the *gradient* based selection. Specifically, a sample with a lower gradient value with the current model parameters is chosen for replacement. Intuitively, the selected sample represents the least significant or a well-learned sample. Experiments are conducted using CIFAR-10, CIFAR-100, and CLEAR-10 benchmarking datasets, and the results are presented in Table 9. We make the following observations. The gradient-based method significantly **increases** the memory population (MP) time due to the frequent gradient computations. As a result, the total training time relatively increases by (i) 86 times for CIFAR-10, (ii) 47 times for CIFAR-100, and (iii) 15 times for CLEAR-10 datasets, respectively, compared to random sample selection without significant performance gains. Based on this empirical evidence, we believe that using gradient based sample selection method will further increase the training time without a significant performance gain. So, we intentionally omit the gradient based method, *online coreset selection* [24], from the baselines.

Table 9: Performance comparison of ECBRS method with random and gradient-based sample selection strategies. We report the arithmetic mean values after repeating each experiment five times independently.

| Dataset | Sample selecton strategy | PR-AUC (A) | PR-AUC (B) | ROC-AUC | MP time (in sec.) | training time (in sec.) | relative increase in total training time |
|---|---|---|---|---|---|---|---|
| CIFAR-10 | random | 0.95 | 0.94 | 0.94 | 0.13 | 61 | |
| | **gradient** | 0.95 | 0.93 | 0.94 | 5190 | 5250 | **86 times** |
| CIFAR-100 | random | 0.640 | 0.640 | 0.636 | 0.118 | 96.1 | |
| | **gradient** | 0.656 | 0.616 | 0.640 | 4523 | 4579 | **47 times** |
| CLEAR-10 | random | 0.890 | 0.885 | 0.887 | 128.2 | 265.9 | |
| | **gradient** | 0.936 | 0.921 | 0.925 | 3770 | 4013 | **15 times** |

**Threshold parameter** ($\gamma$): It indicates the expected number of samples to be present in the buffer memory based on the running statistics (global information). $\gamma(\cdot)$ for each class will depend on the weight $w(\cdot)$ associated with it, and the weight is computed using the negative soft-max over each class running frequency. Here, the ablation study is conducted to understand the implications of the varying $w(\cdot)$ on the performance of the proposed ECBRS on selective benchmarks. The results are present in Table 10, and we make the following observations.

1. $w(\cdot)$ controls the learning attention to the distribution shifts benign and attack classes. We observe that $w(\cdot)$ in the range of 0.1 to 0.9 for NIDS benchmarks (except AnoShift) achieves nearly similar performance on attack data (PR-AUC (A)). However, we have an exception to this premise. Specifically, on the CICIDS-2018 dataset, for $w(\cdot) = 0.3, 0.5$, a performance drop is observed. A similar trend is observed for KDDCUP'99 and NSL-KDD at $w(\cdot) = 0.7$, 0.9, and 0.5. This can be due to the higher value of the $w(\cdot)$ that makes the learner more exposed to the benign data. Based on this study, we observe using the value of 0.1 is always an optimal choice.

2. For AnoShift, in which benign traffic is the minority class, the attack, and benign class performance are low when the $w(\cdot) = 0.1$. However, significant improvement is observed with higher values of $w(\cdot)$.

3. On vision datasets, performance on the benign (PR-AUC (B)) and attack classes (PR-AUC (A)) continues to become stationary.

To conclude, we recommend using a lower value of $w(\cdot)$ for benchmarks with more benign samples and a higher value for benchmarks with more attack traffic.

**Effect of batch size**: Here, we study the effect of varying the batch size on the performance of the proposed ECBRS method, and the results on selective benchmark datasets are presented in Table 11.

Table 10: Effect of the threshold parameter $\gamma(.)$ of the ECBRS on the performance results of the selective benchmark datasets. We report the mean values with each experiment repeated independently four times.

| | NSL-KDD | | | | KDDCUP'99 | | |
|---|---|---|---|---|---|---|---|
| $w(.)$ | PR-AUC (A) | PR-AUC (B) | ROC-AUC | $w(.)$ | PR-AUC (A) | PR-AUC (B) | ROC-AUC |
| 0.1 | 0.949 | 0.963 | 0.967 | 0.1 | 1.000 | 0.993 | 0.999 |
| 0.3 | 0.971 | 0.975 | 0.972 | 0.3 | 1.000 | 0.994 | 0.999 |
| 0.5 | 0.924 | 0.968 | 0.959 | 0.5 | 1.000 | 0.993 | 0.999 |
| 0.7 | 0.968 | 0.972 | 0.970 | 0.7 | 0.948 | 0.938 | 0.942 |
| 0.9 | 0.952 | 0.970 | 0.967 | 0.9 | 0.929 | 0.924 | 0.929 |

| | CICIDS-2018 | | | | AnoShift | | |
|---|---|---|---|---|---|---|---|
| $w(.)$ | PR-AUC (A) | PR-AUC (B) | ROC-AUC | $w(.)$ | PR-AUC (A) | PR-AUC (B) | ROC-AUC |
| 0.1 | 0.999 | 0.999 | 0.999 | 0.1 | 0.880 | 0.868 | 0.790 |
| 0.3 | 0.947 | 0.952 | 0.899 | 0.3 | 0.941 | 0.920 | 0.926 |
| 0.5 | 0.927 | 0.925 | 0.866 | 0.5 | 0.948 | 0.935 | 0.940 |
| 0.7 | 0.998 | 0.998 | 0.998 | 0.7 | 0.948 | 0.938 | 0.942 |
| 0.9 | 0.999 | 0.999 | 0.999 | 0.9 | 0.929 | 0.924 | 0.929 |

| | CIFAR-10 | | | | CLEAR-10 | | |
|---|---|---|---|---|---|---|---|
| $w(.)$ | PR-AUC (A) | PR-AUC (B) | ROC-AUC | $w(.)$ | PR-AUC (A) | PR-AUC (B) | ROC-AUC |
| 0.1 | 0.953 | 0.940 | 0.948 | 0.3 | 0.941 | 0.932 | 0.932 |
| 0.3 | 0.956 | 0.945 | 0.951 | 0.5 | 0.938 | 0.926 | 0.927 |
| 0.7 | 0.945 | 0.934 | 0.941 | 0.7 | 0.942 | 0.932 | 0.933 |
| 0.9 | 0.943 | 0.937 | 0.941 | 0.9 | 0.925 | 0.910 | 0.912 |

| | CIFAR-100 | | | | CLEAR-100 | | |
|---|---|---|---|---|---|---|---|
| $w(.)$ | PR-AUC (A) | PR-AUC (B) | ROC-AUC | $w(.)$ | PR-AUC (A) | PR-AUC (B) | ROC-AUC |
| 0.3 | 0.667 | 0.609 | 0.643 | 0.3 | 0.855 | 0.812 | 0.833 |
| 0.5 | 0.654 | 0.591 | 0.634 | 0.5 | 0.854 | 0.810 | 0.831 |
| 0.7 | 0.661 | 0.592 | 0.637 | 0.7 | 0.853 | 0.814 | 0.831 |
| 0.9 | 0.655 | 0.606 | 0.640 | 0.8 | 0.846 | 0.802 | 0.826 |

On intrusion detection benchmarks like CICIDS-2018, the differences in the performance results with the varying batch size are minimal. On the contrary, the performance measures on the AnoShift dataset are stationary with batch size. We chose 1024 as the batch size for NID benchmarks. We observe a performance degradation on computer vision benchmarks using the batch size 256, particularly on CIFAR-100 and CLEAR-10 datasets. Intuitively, this could be due to the fact that a large batch size leads to a degradation in the quality of the model, which affects the generalization ability [25]. So, we chose 128 as the batch size for vision datasets.

### A.10.2 PAPA

In this section, we study the impact of varying each hyperparameter on the performance of the proposed PAPA method. Specifically, ablation studies are conducted to study the effect of the first task on the performance of PAPA method, memory size ($\mathcal{M}$), and batch size.

**Impact of the different first tasks on PAPA's performance:** The PAPA algorithm uses the MIR algorithm during the first task to generate the error distribution samples. These samples are in turn used to train the GMM. Here, we examine the robustness of the PAPA's performance when the choice of the first task is varied. Towards this, we formulate four different task orders with varying first tasks. We selectively conducted this study on the large-size intrusion detection and vision benchmarks with distribution shifts over time. To understand the bias of the first task on the learning of the PAPA algorithm on remaining tasks, *variance* in the performance results over selective benchmarks is computed and the results are present in Table 12. The variance is low (in the range of 0 to $10^{-4}$) across all the benchmarks, thereby empirically signifying the task order robustness of the proposed PAPA algorithm. More detailed performance results on different task orders are available in Table 13.

Table 11: Effect of the batch size on the performance of the proposed ECBRS method using selective benchmark datasets. We report the mean values with each experiment repeated independently four times.

| | CICIDS-2018 | | | | AnoShift | | |
|---|---|---|---|---|---|---|---|
| Batch size | PR-AUC (A) | PR-AUC (B) | ROC-AUC | Batch size | PR-AUC (A) | PR-AUC (B) | ROC-AUC |
| 256 | 0.862 | 0.869 | 0.740 | 256 | 0.948 | 0.939 | 0.945 |
| 512 | 0.947 | 0.852 | 0.904 | 512 | 0.944 | 0.933 | 0.937 |
| 1024 | 0.999 | 0.999 | 0.999 | 1024 | 0.949 | 0.944 | 0.948 |
| 2048 | 1.000 | 1.000 | 0.999 | 2048 | 0.945 | 0.939 | 0.945 |

| | CIFAR-10 | | | | CIFAR-100 | | |
|---|---|---|---|---|---|---|---|
| Batch size | PR-AUC (A) | PR-AUC (B) | ROC-AUC | Batch size | PR-AUC (A) | PR-AUC (B) | ROC-AUC |
| 16 | 0.948 | 0.944 | 0.948 | 16 | 0.640 | 0.600 | 0.638 |
| 32 | 0.955 | 0.947 | 0.951 | 32 | 0.666 | 0.613 | 0.649 |
| 64 | 0.952 | 0.941 | 0.947 | 64 | 0.671 | 0.603 | 0.646 |
| 128 | 0.953 | 0.941 | 0.948 | 128 | 0.663 | 0.611 | 0.663 |
| 256 | 0.953 | 0.940 | 0.949 | 256 | 0.646 | 0.582 | 0.622 |

| | CLEAR-10 | | | | CLEAR-100 | | |
|---|---|---|---|---|---|---|---|
| Batch size | PR-AUC (A) | PR-AUC (B) | ROC-AUC | Batch size | PR-AUC (A) | PR-AUC (B) | ROC-AUC |
| 16 | 0.955 | 0.944 | 0.947 | 16 | 0.861 | 0.815 | 0.838 |
| 32 | 0.953 | 0.933 | 0.942 | 32 | 0.860 | 0.804 | 0.834 |
| 64 | 0.961 | 0.941 | 0.950 | 64 | 0.853 | 0.806 | 0.828 |
| 128 | 0.937 | 0.926 | 0.926 | 128 | 0.854 | 0.807 | 0.831 |
| 256 | 0.897 | 0.905 | 0.896 | 256 | 0.860 | 0.829 | 0.845 |

Table 12: Variance in the performance results of the proposed PAPA algorithm with different first tasks. We conduct each experiment four times independently, and sample variance is reported.

| Dataset | PR-AUC (A) | PR-AUC (B) | ROC-AUC |
|---|---|---|---|
| CICIDS-2018 | 0 | 0 | 0 |
| AnoShift | $10^{-4}$ | $2.6 \times 10^{-4}$ | $2 \times 10^{-4}$ |
| CIFAR-10 | $8.3 \times 10^{-5}$ | $2 \times 10^{-5}$ | $3.3 \times 10^{-5}$ |
| CLEAR-10 | $2.3 \times 10^{-5}$ | $3.9 \times 10^{-5}$ | $2.8 \times 10^{-5}$ |
| CIFAR-100 | $7.8 \times 10^{-4}$ | $2.0 \times 10^{-4}$ | $3.2 \times 10^{-4}$ |
| CLEAR-100 | $2.5 \times 10^{-5}$ | $4.6 \times 10^{-5}$ | $4.9 \times 10^{-5}$ |

**Effect of memory size($\mathcal{M}$) on PAPA performance:** Here, we study the impact of the buffer memory size on the proposed PAPA algorithm using selective network intrusion detection and computer vision benchmark datasets. Performance results are represented in Table 14. On intrusion detection benchmark AnoShift, performance results steadily increase until $\mathcal{M} = 5000$. Later, an increase in the performance is observed at $\mathcal{M} = 10000$, and the subsequent performance results become stationary after that. On CICIDS-2018, a sudden increase in the performance is observed at $\mathcal{M} = 5000$. Intuitively, this indicates that a specific $\mathcal{M}$ value may exist for each dataset, after which a sudden increase in the performance results is observed, and performance results on subsequent $\mathcal{M}$ values become stationary or may increase. The performance results on the vision are slightly varied until a specific value $\mathcal{M}$. After that, an increase is observed and becomes steady, an analogy identical to the NIDS benchmarks. Specifically, these $\mathcal{M}$ values 500, 666, 500, and 1000 for CIFAR-10, CLEAR-10, CIFAR-100, and CLEAR-100, respectively. The training set sizes of the CIFAR-100 and CIFAR-10 are equal, so we use a memory size of 500, and for CLEAR-10, we use 666. Eventually, for CLEAR-100, a memory size of 2666 is used.

**Effect of batch size on PAPA performance**: Here, we study the impact of the batch size on the performance of the proposed PAPA algorithm on the selective intrusion detection and vision benchmark datasets, and the results are present in Table 15. Performance results are low on the

Table 13: Performance comparison of the proposed PAPA algorithm with various task orders on selective benchmark datasets. Each experiment is repeated four times independently, and mean values are reported.

| | CICIDS-2018 | | | AnoShift | | |
|---|---|---|---|---|---|---|
| Task order index | PR-AUC (A) | PR-AUC (B) | ROC-AUC | PR-AUC (A) | PR-AUC (B) | ROC-AUC |
| 1 | 0.999 | 0.999 | 0.999 | 0.938 | 0.915 | 0.923 |
| 2 | 0.999 | 0.999 | 0.999 | 0.953 | 0.937 | 0.945 |
| 3 | 0.999 | 0.999 | 0.999 | 0.952 | 0.933 | 0.938 |
| 4 | 0.999 | 0.999 | 0.999 | 0.933 | 0.902 | 0.913 |
| | CIFAR-10 | | | CLEAR-10 | | |
| Task order index | PR-AUC (A) | PR-AUC (B) | ROC-AUC | PR-AUC (A) | PR-AUC (B) | ROC-AUC |
| 1 | 0.940 | 0.946 | 0.944 | 0.949 | 0.931 | 0.941 |
| 2 | 0.946 | 0.950 | 0.946 | 0.946 | 0.926 | 0.935 |
| 3 | 0.960 | 0.953 | 0.957 | 0.948 | 0.931 | 0.937 |
| 4 | 0.956 | 0.943 | 0.951 | 0.957 | 0.941 | 0.947 |
| | CIFAR-100 | | | CLEAR-100 | | |
| Task order index | PR-AUC (A) | PR-AUC (B) | ROC-AUC | PR-AUC (A) | PR-AUC (B) | ROC-AUC |
| 1 | 0.672 | 0.677 | 0.674 | 0.846 | 0.792 | 0.823 |
| 2 | 0.707 | 0.662 | 0.684 | 0.846 | 0.796 | 0.826 |
| 3 | 0.663 | 0.648 | 0.658 | 0.839 | 0.780 | 0.813 |
| 4 | 0.722 | 0.679 | 0.701 | 0.836 | 0.788 | 0.812 |

Table 14: Evaluating the effect of different buffer memory sizes on the proposed PAPA algorithm using various benchmark datasets. Each experiment is repeated four times independently, and mean values are reported.

| | CICIDS-2018 | | | | AnoShift | | |
|---|---|---|---|---|---|---|---|
| $\mathcal{M}$ | PR-AUC (A) | PR-AUC (B) | ROC-AUC | $\mathcal{M}$ | PR-AUC (A) | PR-AUC (B) | ROC-AUC |
| 2000 | 0.940 | 0.946 | 0.892 | 2000 | 0.907 | 0.918 | 0.916 |
| 5000 | 0.996 | 0.995 | 0.994 | 5000 | 0.924 | 0.929 | 0.927 |
| 10000 | 0.999 | 0.999 | 0.999 | 10000 | 0.945 | 0.932 | 0.937 |
| 20000 | 0.999 | 0.999 | 1.000 | 20000 | 0.947 | 0.932 | 0.940 |
| 40000 | 0.999 | 0.999 | 0.998 | 40000 | 0.944 | 0.925 | 0.932 |
| | CIFAR-10 | | | | CLEAR-10 | | |
| $\mathcal{M}$ | PR-AUC (A) | PR-AUC (B) | ROC-AUC | $\mathcal{M}$ | PR-AUC (A) | PR-AUC (B) | ROC-AUC |
| 50 | 0.929 | 0.923 | 0.928 | 50 | 0.946 | 0.934 | 0.938 |
| 250 | 0.921 | 0.919 | 0.923 | 250 | 0.953 | 0.937 | 0.945 |
| 500 | 0.948 | 0.936 | 0.944 | 666 | 0.943 | 0.927 | 0.932 |
| 1000 | 0.945 | 0.926 | 0.943 | 1000 | 0.942 | 0.934 | 0.934 |
| 2000 | 0.953 | 0.949 | 0.951 | 1200 | 0.940 | 0.926 | 0.930 |
| | CIFAR-100 | | | | CLEAR-100 | | |
| $\mathcal{M}$ | PR-AUC (A) | PR-AUC (B) | ROC-AUC | $\mathcal{M}$ | PR-AUC (A) | PR-AUC (B) | ROC-AUC |
| 50 | 0.669 | 0.661 | 0.662 | 250 | 0.860 | 0.810 | 0.839 |
| 250 | 0.642 | 0.638 | 0.643 | 500 | 0.860 | 0.812 | 0.837 |
| 500 | 0.673 | 0.647 | 0.672 | 1000 | 0.830 | 0.788 | 0.812 |
| 1000 | 0.667 | 0.628 | 0.637 | 2666 | 0.845 | 0.793 | 0.823 |

CICIDS-2018 dataset on the batch size of 512; thereafter, it increases with batch size and becomes stable. On the AnoShift benchmark, attack data traffic detection (PR-AUC (A)) performance values are stable with increasing batch size, whereas benign data traffic detection (PR-AUC (A)) performance values are unstable with increasing batch size. PR-AUC (A) values increase from batch size 64 on

Table 15: Effect of the batch size on the performance of the proposed PAPA method using selective benchmark datasets. We report the mean values with each experiment repeated independently four times.

| CICIDS-2018 | | | | AnoShift | | | |
|---|---|---|---|---|---|---|---|
| Batch size | PR-AUC (A) | PR-AUC (B) | ROC-AUC | Batch size | PR-AUC (A) | PR-AUC (B) | ROC-AUC |
| 256 | 0.841 | 0.850 | 0.697 | 256 | 0.943 | 0.938 | 0.946 |
| 512 | 0.996 | 0.995 | 0.994 | 512 | 0.945 | 0.929 | 0.937 |
| 1024 | 0.999 | 0.999 | 0.999 | 1024 | 0.945 | 0.932 | 0.937 |
| 2048 | 0.999 | 1.000 | 0.999 | 2048 | 0.944 | 0.935 | 0.938 |

| CIFAR-10 | | | | CIFAR-100 | | | |
|---|---|---|---|---|---|---|---|
| Batch size | PR-AUC (A) | PR-AUC (B) | ROC-AUC | Batch size | PR-AUC (A) | PR-AUC (B) | ROC-AUC |
| 32 | 0.927 | 0.933 | 0.933 | 32 | 0.689 | 0.669 | 0.686 |
| 64 | 0.947 | 0.937 | 0.945 | 64 | 0.685 | 0.653 | 0.675 |
| 128 | 0.948 | 0.936 | 0.944 | 128 | 0.673 | 0.647 | 0.672 |
| 256 | 0.945 | 0.927 | 0.939 | 256 | 0.675 | 0.646 | 0.667 |
| 512 | 0.945 | 0.938 | 0.942 | 512 | 0.688 | 0.688 | 0.679 |

| CLEAR-10 | | | | CLEAR-100 | | | |
|---|---|---|---|---|---|---|---|
| Batch size | PR-AUC (A) | PR-AUC (B) | ROC-AUC | Batch size | PR-AUC (A) | PR-AUC (B) | ROC-AUC |
| 16 | 0.951 | 0.943 | 0.945 | 16 | 0.817 | 0.768 | 0.798 |
| 32 | 0.967 | 0.950 | 0.959 | 32 | 0.844 | 0.786 | 0.822 |
| 64 | 0.963 | 0.944 | 0.953 | 64 | 0.839 | 0.785 | 0.816 |
| 128 | 0.943 | 0.927 | 0.932 | 128 | 0.845 | 0.793 | |
| 256 | 0.928 | 0.918 | 0.917 | 256 | 0.855 | 0.821 | 0.843 |

CIFAR-10. On CIFAR-100, reduced performance is observed for the batch sizes of 128 and 256. For CLEAR-10 and CLEAR-100, performance results are low for the batch size 16. Specifically, for CLEAR-10, a performance drop is observed for batch sizes of 128 and 256. However, the performance is stationary for all the batch sizes greater than 16 for the CLEAR-100 dataset.

### A.11  Implementation details and hyperparameter selection

We conducted experiments on the multi-layer perceptron (MLP) for all intrusion detection benchmarks and ResNet-18 (pre-train on imageNet1k) architecture for SVHN, CIFAR-10/100, and CLEAR-10/100 datasets. Each benchmark dataset is split into three parts: 75% for train, 23% for testing, and 2% for validation. For each experiment, the number of epochs is set to five. We use open source continual learning library *avalanche* (version 0.2.1) [26] implementation for baseline methods like EWC, SI, GEM, A-GEM, and GSS-greedy. However, direct usage of these implementations has certain technical difficulties, especially for IDS benchmarks. So, we tailor them to work on the benchmarks. However, during this process, we faced a lot of challenges. One such challenge encountered during the MLP training on the CICIDS2018, CTU-13 benchmark using *gss-greedy* and *agem* algorithm. After much debugging, we found it was the issue with the gradients. Eventually, training is resumed by normalizing or replacing the nan gradient. We implemented MIR, CBRS, and ECBRS using PyTorch library [27]. We selectively chosen the buffer memory size for each dataset based on the number of training samples in the respective datasets.

**System/Hardware details** We ran our experiments on the system with the following specifications: Operating system-Ubuntu 18.04.6 LTS, memory-376 GB, number of cores-104 (Intel(R) Xeon(R) Gold 6230R CPU @ 2.10GHz), and 2 Nvidia Quadro RTX 5000 GPU.

**Reproducibility:** The complete codebase and train data files used in this work are kept at this link.

List of values used for hyperparameters per each dataset are presented in Table 16, and ablation studies on the hyperparameters are discussed in subsection A.10 . Replay size is the number of samples selected from the buffer memory for replay. #tasks is the number of task per dataset. Pattern per experience (PPE) (an avalanche library convention) is the number of samples per task stored in the buffer memory. FC indicates a fully connected multi-layer perceptron.

Table 16: Hyperparameter details of each network intrusion detection and computer vision benchmark datasets.

| Dataset | $\mathcal{M}$ | Replay size | #tasks | Batch size | PPE | input size | Architecture |
|---|---|---|---|---|---|---|---|
| NSL-KDD | 1333 | 1000 | 5 | 500 | 200 | 38 | FC:100,500,250,50,1 |
| KDDCUP'99 | 5333 | 4000 | 5 | 1024 | 800 | 38 | FC:100,500,250,50,1 |
| CICIDS-2017 | 13334 | 10000 | 10 | 1024 | 100 | 51 | FC:100,250,50,1 |
| CICIDS-2018 | 13334 | 10000 | 10 | 1024 | 100 | 51 | FC:100,500,250,50,1 |
| UNSW-NB15 | 6666 | 5000 | 9 | 1024 | 500 | 202 | FC:100,250,500,150,50,1 |
| CTU-13 | 1500 | 1000 | 5 | 1024 | 1000 | 39 | FC:100,250,50,1 |
| AnoShift | 13333 | 10000 | 10 | 1024 | 1000 | 18 | FC:100,500,250,50,1 |
| SVHN | 500 | 375 | 9 | 128 | 50 | $3 \times 32 \times 32$ | ResNet-18,1 |
| CIFAR-10 | 500 | 375 | 9 | 128 | 50 | $3 \times 32 \times 32$ | ResNet-18,FC:100,50,1 |
| CIFAR-100 | 500 | 375 | 19 | 128 | 50 | $3 \times 32 \times 32$ | ResNet-18,FC:100,50,1 |
| CLEAR-10 | 666 | 500 | 10 | 128 | 50 | $3 \times 224 \times 224$ | ResNet-18,FC:100,50,1 |
| CLEAR-100 | 2666 | 2000 | 10 | 128 | 50 | $3 \times 224 \times 224$ | ResNet-18,FC:100,50,1 |

## A.12 Occurrence of task dissimilarity between two different tasks is rare

We will demonstrate that the error distributions between different tasks remain similar even with varying task orders, and it can be modeled using two-component GMM. Additionally, we will illustrate cases where the dissimilarity in the error distribution occurs due to two dissimilar tasks, which is rare in the context of network intrusion detection systems.

**A two-component GMM is capable enough for various task orders**: We direct the reader's attention to A.10.2 and Tables 12, 13 of this supplementary material. In these sections, we conducted experiments using various datasets with four distinct task orders, each having a different first task. The performance results for each task order are presented in Tables 12, 13 which illustrate the variance in these reported values. Notably, we observe that the variance of performance results falls within the range of $0$ to $10^{-4}$. This observation indicates that the influence of the different first tasks used in the computation of the Gaussian Mixture Model (GMM) on the performance of PAPA is minimal. This underscores that the GMM constructed based on the initial error distribution (ED) effectively approximates the ED of subsequent tasks, irrespective of the choice of the first task.

**Why the occurrence of two dissimilar tasks in a NIDS setting is rare?**: To prove our premise, We designed an experiment to learn tasks sequentially from MNIST and CIFAR-10. Before initiating the experiments, each task's mean OTDD value, considering its relation to the remaining tasks, we call it relative OTDD ($OTDD_{rel}$). We want to clarify that the $OTDD_{rel}$ values reported are related to the MNIST+CIFAR-10 experiment and are not computed between two adjacent tasks. Instead, for a task (say $t_1$), the $OTDD_{rel}$ value is computed with other remaining tasks ($t_2,t_3,...,t_9$) as follows

$$OTDD_{rel}(t_1) = \frac{1}{8} \sum_{i=2}^{9} OTDD(t_1, t_i)$$

However, these values (shown in Table 18) are not uniformly distributed compared to the experiments conducted solely using the CIFAR-10 dataset. This discrepancy suggests a higher dissimilarity within these experiments.

Table 17: Relative optimal transport dataset distance values for each task in CIFAR-10 and MNIST+CIFAR-10 experiments.

| Dataset | Task1 | Task2 | Task3 | Task4 | Task5 | Task6 | Task7 | Task8 | Task9 | Avg. $OTDD_{rel}$ values |
|---|---|---|---|---|---|---|---|---|---|---|
| CIFAR-10 | 258 | 255 | 220 | 243 | 212 | 241 | 225 | 236 | 244 | 237.22 $\pm$10.06 |
| MNIST+CIFAR-10 | 310 | 448 | 310 | 444 | 310 | 456 | 310 | 470 | 310 | 374.17$\pm$50.02 |

Furthermore, during these experiments, we observed a significant disparity in the performance values between the MIR and PAPA algorithms. Notably, all these experiments are performed using 20 distinct task orders, and the reported results encompass both mean and standard deviation values.

Table 18: Performance results of the MNIST+CIFAR-10 experiments

| Algorithm | PR-AUC (A) | PR-AUC (B) | ROC-AUC |
|-----------|-----------|-----------|---------|
| MIR | 0.675±0.053 | 0.700±0.030 | 0.657±0.040 |
| PAPA | 0.645±0.047 | 0.661±0.099 | 0.628±0.080 |

Additionally, we aimed to validate this observation through an empirical approach. In this pursuit, after conducting experiments using the MIR algorithm, we calculated error values for the parameters and tried to fit the Gaussian Kernel Density Estimator (KDE). However, we encountered a singular matrix error while attempting the KDE fit, preventing us from generating a Gaussian approximation. It is noteworthy that this issue does not arise in other experimental scenarios. From this experience, we can deduce that the two-component GMM is feasible whenever the variance in the mean OTDD value of each task concerning the remaining tasks is low.

In the MNIST+CIFAR-10 experiment, we explicitly show $OTDD_{rel}$ values to showcase an **extreme case** where the proposed PAPA method would not perform well. However, in practice, finding these values is only possible if access to past and future tasks is allowed, but such access is restricted in the continual learning setting. On the other hand, computing the OTDD values between two adjacent tasks will not help determine the higher similarity between the two tasks (refer to Table 19).

Table 19: Optimal transport dataset distance values computed between adjacent tasks. The average OTDD value is computed as the mean value over all the OTDD values.

| Dataset | (1,2) | (2,3) | (3,4) | (4,5) | (5,6) | (6,7) | (7,8) | (8,9) | Avg. OTDD values |
|---------|-------|-------|-------|-------|-------|-------|-------|-------|------------------|
| MNIST+CIFAR-10 | 424.29 | 404.25 | 405.68 | 405.70 | 409.09 | 409.09 | 407.59 | 407.59 | 409.16±4.399 |

The reason for calling the MNIST+CIFAR-10 experiment an extreme case is based on the following two observations.

1. In this experiment, training data arrives from two different data sources in the domain incremental setting with differing characteristics (size, image type, etc.; refer to Table 20). However, in practice, finding a use case where the experiments with such a shift between tasks with different characteristics (switching between grayscale and RGB images learning sequentially) is difficult.

Table 20: Characteristics of the MNIST and CIFAR-10 datasets

| Dataset | Size | No of channels | Image type |
|---------|------|----------------|------------|
| MNIST | 28 X 28 | 1 | Grayscale |
| CIFAR-10 | 32 X 32 | 3 | RGB |

2. The AnoShift dataset contains diverse traffic from five different networks of Kyoto University spanning over ten years. This is the challenging dataset in network intrusion detection experiments that contains data points from different networks compared to other datasets (created over a short time on a single network; refer to Table 3 for mean OTDD values in each dataset, in which OTDD value is computed between adjacent tasks). For example, consider datasets like CICIDS2017 and CICIDS2018, where datasets were framed using the traffic from a single network in which a high distribution shift may not happen. On the other hand, our proposed PAPA on the challenging AnoShift dataset (with diverse traffic) works better than baseline methods.

To conclude, our intention in framing MNIST+CIFAR-10 experiments is to showcase an extreme case where the proposed approach is expected to not perform well in domain incremental learning. However, encountering such a scenario is rare in the NIDS setting.

### A.13 Limitations and broader impact

This work is motivated by two of our empirical observations regarding the class imbalance in the large-size benchmarks and the scalability in an online task-free continual learning setting for network intrusion detection. We validate our contributions on network intrusion and computer vision benchmark datasets using standard multi-layer perceptron and ResNet-18 architectures. Based on our empirical observations, we categorize the proposed approach's *limitations* into two groups. They are (i) training-related and (ii) practical constraints of the proposed approaches.

**Training related**: The limitations outlined here pertain to the training procedures of the proposed methodologies. These include a lack of real-world datasets for validation, hardware requirements, and possible remedies to these limitations.

1. **Public datasets:** Similar to numerous prior studies in Network Intrusion Detection (NID), proposed ECBRS and PAPA are evaluated using publicly accessible NID datasets. Most of these datasets are simulated in a controlled environment, which may not accurately reflect real-world network traffic. As a result, the gap becomes wider between security practitioners and researchers. One solution to this issue is to create a real-world dataset that adheres to a set of standardized guidelines established by cybersecurity practitioners.

2. **Hardware**: Our work uses gradient-based optimizers like stochastic gradient descent that require better hardware specifications like GPU to accelerate the training process compared to shallow methods like random forest. However, in real-world deployment, we believe MLOps could support our system due to low latency and separate environments for inference and training. MLOps platforms' robust hardware minimizes model complexity's effect on inference.

**Practical considerations:** The practical limitations range from tackling open-world settings, explainability, etc.

1. **Close world setting:** The proposed model tries to learn the distribution shifts in the benign and attack classes to avoid catastrophic forgetting, which could lead to closed-world learning. This is our initial attempt to understand two interdisciplinary areas: NID and continual learning, so we are considering an open-world setting. To tackle an open-world setting, whenever new attack traffic arrives, and the model is unsure of its inference label, we could then bring security analysts into the loop to help improve the detection capabilities of the model.

2. **Labeling assumption**: We assume fully labeled data is available in this work. However, the actual developer of machine learning-based NIDS may not have access to fully labeled data. In future work, we will address the NIDS problem from a semi-supervised or unsupervised continual learning perspective.

3. **Explainability:** Generating alerts (to the security operation center) in response to known/novel attack traffic is the primary responsibility of the NIDS. The $explainability$ of the reason for generating alerts is beneficial to security analysts. We will consider explainability in our future research efforts.

4. **Relating to other domains:** To make our work more tangible to other machine learning-related fields like anomaly detection, fraud detection, and medical image anomaly detection, more validation is required on related datasets and different architectures to understand and address the domain-specific challenges.

**Broader Impact**: NIDS is a crucial element of the cybersecurity toolkit, playing a critical role in safeguarding ICT infrastructure against the ever-growing threat of cyber-attacks. Learning-based NIDS (L-NIDS) has emerged as a popular topic in academic research within the security domain. However, despite its success in academia, the practical implementation of L-NIDS in real-world operational deployments faces several challenges. One such challenge involves adapting to the continuous changes in network traffic distribution. Our work tackles this problem by approaching intrusion detection as a supervised binary classification task. As with any other research endeavor, our work has both advantages and limitations. Besides its relevance to intrusion detection, this work holds broader implications for various fields, including anomaly detection and fraud detection. It is important to note that any potential negative consequences arising from our work, such as legal

and ethical concerns, are not unique to our research specifically but rather common considerations associated with any new developments in the field of machine learning as a whole.

### A.14 Class balanced reservoir sampling (CBRS) [23]

Let's begin by introducing our notation. In the context of a memory with a size 'm,' it is defined as *filled* when all of its 'm' storage units are in use. If a particular class contains the highest number of instances among all the different classes present in the memory, such a class is referred to as the *largest* class. It's important to note that two or more classes can share the title of *largest* if they are both equal in size and the most numerous. Furthermore, a class is designated as *full* if it is either currently the largest class or has held this status in previous time steps. Once a class achieves *full* status, it retains this designation in the future. These classes are termed *full* because CBRS restricts their ability to expand further in size.

The CBRS sampling scheme can be divided into two phases. In the initial phase, all incoming stream instances are stored in memory as long as the memory is not yet full. We transition to the second phase once the memory reaches its capacity and becomes filled. When a stream instance $(x_i, y_i)$ is received during the second phase, the algorithm initially checks whether $y_i$ is associated with a full class. If it is not, the incoming instance takes the place of another instance belonging to the largest class. This aspect of CBRS serves to address class imbalances within the memory. Conversely, if the received instance corresponds to a class **c**, it replaces a randomly chosen stored instance of the same class **c** with a probability of $m_c/n_c$. Here, $m_c$ represents the number of instances from class c currently stored in memory, and $n_c$ signifies the total number of stream instances encountered from class c up to that point. We outline the pseudocode for the CBRS algorithm in Algorithm 1.

---

**Algorithm 1** Memory population for CBRS

---

    **Input:** data stream: $(x_i, y_i)_{i=1}^n$
   **for** $i = 1$ **to** $n$ **do**
     **if** memory is **not** filled **then**
       store $(x_i, y_i)$
     **else**
       **if** $c \equiv y_i$ is **not** a full class **then**
         find all the instances of the largest class select from them an instance at random overwrite the selected instance with $(x_i, y_i)$
       **else**
         $m_c \leftarrow$ number of currently stored instances of class ($c \equiv y_i$)
         $n_c \leftarrow$ number of stream instances of class $c \equiv y_i$ encountered thus far:
         sample u $\sim$ Uniform(0,1)
         **if** u $\leq m_c/n_c$ **then**
           pick a stored instance of class $c \equiv y_i$ at random and replace it with $(x_i, y_i)$
         **else**
           Ignore $(x_i, y_i)$
         **end if**
       **end if**
     **end if**
   **end for**

---