# OpenReview forum: "Augmented Memory Replay-based Continual Learning Approaches for Network Intrusion Detection"
_NeurIPS.cc/2023/Conference — NeurIPS 2023 poster_

### Official Review · Reviewer_ZBnA · 2023-06-19

**Soundness:** 3 good
**Presentation:** 3 good
**Contribution:** 3 good
**Rating:** 6
**Confidence:** 5

**Summary:**

The paper considers the application of deep learning (DL) for network intrusion detection systems (NIDS). The paper correctly points out that, in real contexts, NIDS must be continously updated with new data-points to mitigate the impact of "concept drift". A way to do so is by employing "continual learning" (CL) methods---which, unfortunately, are affected by many issues which the paper seeks to address.
Specifically, the main contribution is a novel approach to deal with (i) class-imbalance problems leading to "catatrophic forgetting" (CF); and (ii) the computational overhead deriving from ways to mitigate CF. These contributions, whose intuitions are based on empirical evidence, are experimentally assessed on many datasets. The results show a good improvement over the baselines.

**Strengths:**

+ Relevant problem
+ Large evaluation
+ Good results
+ (somewhat) theoretically grounded
+ Multiple repetitions
+ Good quality of writing
+ The supplementary material is rich with details

The paper addresses a relevant problem within the machine learning (ML) domain, which has received only limited attention in the specific context of network intrusion detection (NID). The proposed methods are theoretically grounded (especially Section 3.1), and the findings are derived by a large sets of experiments carried out on various datasets (which are not limited to NID data) and considering many baselines. The conclusions are drawn by repeating the experiments 5 times, increasing the overall soundness of the results---showing a substantial improvements over the baselines.

**Weaknesses:**


## High Level

- Inappropriate datasets
- (somewhat) confusing theoretical arguments
- Bold assumptions
- (risk of) data snooping
- (some) missing details

## Low Level (and suggestions)

Below is an extended description of the abovementioned weaknesses. I will also provide some actionable means to rectify such issues, as well as abundant references that can be used to improve this paper and/or support some of my critiques.

### Inappropriate datasets

This is by far the biggest concern I have with the paper: the data used as basis for the experiments is inappropriate to test the hypothesis and provide a _convincing_ answer. Let me explain.

First, the paper deals with the problem of NID. As such, _any finding that derives on data that does not pertain to NID is redundant_. This automatically makes all the evaluations carried out on, e.g., on CIFAR, SVHN, CLEAR to be of zero-value to the NID community (do note that a recent paper [D] highlighted that many papers on security problems carry out evaluations on data that does not reflect a security context, leading to skepticism by practitioners). I acknowledge that these datasets were added due to some limitations of (some) existing datasets for NID, but this is not an acceptable reason: if the authors were better aware of the NID context, they would know that there are (publicly available) datasets that allow a better representation of a real CL scenario. (more on this below).

Second, among the chosen NID datasets, there is the NSL-KDD and KDD Cup'99. These datasets are well-known to be flawed [C]; furthermore, they are also almost 25 years old, and the security community does not find them to be of any interest from a practical viewpoint [B, I]. Hence, even these experiments do not allow to provide a convincing argument in favor of the paper's conclusions.

Third, the NID datasets also include the CIC-IDS17 and its enhanced variant, the CIC-IDS18. Unfortunately, also these two datasets are flawed [G, H]. Note that [G] came out in 2021, and it has already been well-received by the NID community (e.g., [A, F]), so it is concerning that this paper (which has been submitted to a top-venue such as NeurIPS) performs the experiments on the "flawed" variant of these two datasets---especially given that a "fixed" version exists (provided both in [G] and [H]). In short, these experiments are questionable; plus, what is even more questionable is that CIC-IDS17 and CIC-IDS18 are used to derive some observations that motivate the theoretical design of the proposed method.

Fourth, overlooking the previous two points, all the NID datasets span over a very short period of time, and do not enable any assessment that can be used to test "time-aware" applications of machine learning-based NIDS [F] -- or, at least, do so in a way that is appropriate for NeurIPS. The only exception could, potentially, be ANONIDS: however, such a dataset contains data-points from different networks, which raises many concerns [S, O]. Hence, even this dataset has dubious utility for the problem at hand.
The way to circumvent all these issues is not to use "image datasets", but rather to use NID datasets captured (i) over a long period of time and (ii) in the same network: _such datasets **exist**_, and a prominent example is the MCP dataset, which is built upon the well-known CTU13 dataset (see [J,K]).

To summarize: all the experiments carried out in the paper are performed on datasets that are inappropriate to test the underlying hypothesis without there being a sensible reason to do so.

### Bold Assumptions

This is a "pragmatic" weakness, which does not invalidate the paper, but significantly limits its real-world relevance.

Put simply, the paper proposes a method that is rooted in the application of "deep learning" (DL) for NID. The problem, however, is that real developers of ML-NIDS are very skeptical of DL in NID [L]. The reasons are many, but at the basis of this is that they are hardly explainable [M], but also that they are outperformed by "traditional" machine learning approaches (assuming that they can be applied). For example, [E] shows that multi-layer perceptrons are inferior to a random forest from a detection-performance perspective. Moreover, [B] reveals that fine-tuning a DL method requires months of data. Finally, the recent [F] shows that training "shallow" methods (such as decision trees) is very short. Note that [F] uses similar datasets as the ones used in this paper: to give some context, training a binary classifier using a decision tree on UNSW-NB15 requires less than 5s, whereas the MLP used in this paper requires 350s (looking at Table 3) -- and the experiments done in this paper entail much better hardware.

In light of this, it is questionable whether the proposed method has any relevance in reality. Yes, perhaps it provides some advantages -- but if the price of such advantages is an unacceptably high training time (which the method itself seeks to reduce) then the overall contribution of this paper to the state-of-the-art is low.

To address this issue, I invite the authors to include also "shallow" methods that do not entail (deep) neural networks, and show that the proposed method allows the resulting architectures to achieve a comparable degree of performance of "shallow" methods.


Notwithstanding, there is another "bold assumption" made in the paper: the fact that the "malicious classes" are going to remain stable. Indeed (to the best of my understanding), the paper seeks to analyze the effect of distribution shift of the "benign" samples (this is explicitly stated in the introduction), and the set of malicious classes is always known beforehand. This is quite unrealistic (and it had been known since [O]), and may further decrease the value of the proposed method in practice: what would happen in case a new "unknown" attack appears, and is then assigned a "new" label?

On this note, the other "practical" limitation is that the paper assumes that there will be an influx of (correctly labelled) samples that are fed into the system. How is this done in practice? Abundant work (see [N] for a summary) pointed out that doing this is expensive, and especially so in NID context. As a matter of fact, what is not shown in the paper is a clear "use-case" that depicts how the system is designed to be deployed in practice: without such a schematic, it is difficult to determine if the proposed method is even "conceivable" to be deployed into real systems (as also highlighted in [F], practitioners are very interested in the "system infrastructure" envisioned in research papers). Note that I did look in the supplementary material, but the "schematic" provided in Figure 3 does not allow to answer my doubts: the "continual learning module" appears to be a "black-box" that receives its inputs from a "training dataset". However, how is such training data collected (given that we are in an "continual learning" setting)? The way I see it, it requires an enormous amount of manual effort.


### (somewhat) confusing technical arguments

While I had no issues in understanding the rationale of Section 3.1 (aside from the following unclear sentence ```In our approach, we choose this parameter based on global information.```), I found that the description of Section 3.2 to be lacking in terms of clarity and soundness. Setting aside that the experiments are based on "inappropriate" datasets, I did not find compelling evidence of ```virtual SGD parameter update is a slowly varying process.``` -- and this may be due to Figure 3 not being introduced in any way (e.g., what do the x and y axis report? what is the line? the y-axis is also not uniform).

I also did not find the "motivation" (mentioned in the Introduction) that leads up to Section 3.2 to be clear. To my understanding, the problem is that the implementation of memory-replay techniques is expensive from a computational perspective---but is this really the case for NID? As I mentioned above, "shallow" methods are much faster to train.

Also, at the end of the Introduction it stated that ```[PAPA] lead[s] to improved scalability.```. What does this mean? The key-term is "scalability". I do not see any evidence in Section 3.2 that makes me believe that PAPA leads to better scalability. I invite reading [P, Q].


### Potential Data Snooping

I have reason to believe that the methodology followed in the paper is affected by the "data snooping" problem [15]. In other words: the proposed method is designed in a way that would not be typically "known" in advance.

This is epitomized by Section 3.2, wherein some conclusions are drawn by analyzing CIC-IDS18 (and CIFAR), and then used to develop the proposed methodology. At the same time, this is also likely to be present in the experimental setup:

> The intuition behind the memory size is to store nearly 1% of the total training sample of the benchmark dataset in the buffer memory, and 75% of the buffer memory samples will be used for replay.

The point is that a real developer has no clue about the size of the "training dataset": in real settings, an organization will collect some training data, and then use it to train a model; however, they would not know "how much data will appear in the future". It is possible to address this later issue by assuming a "fixed amount" (e.g., 1000 samples) which does not depend on the size og a given benchmark (this is what is done, e.g., in [N]).


### Some additional issues:

* In the caption of Figure 1, it is not stated what "m" refers to
* Bad formatting on Page 6 (the text overlaps with Algorithm 2)
* Also about formatting: tables and figures (and their captions) are almost impossible to discern from the text itself. I think the authors played too much with margins.
* Given the importance of hardware in determining the runtime [F], the main paper should include also details on the experimental platform.
* There is an excessive usage of acronyms in the paper. For instance, SBCP occurs only 3 times (and is strikingly similar to CBRS). Also, MR and MIR are very similar in how they appear, but they denote different concepts. HIDS is also redundant.
* Reference [14] is incomplete
* In Section 3.1, I had a hard time distinguishing between terms such as "maximal", "majority", "full", "largest."
* The "Limitations" are only mentioned in the supplementary material
* The features used to train the models are not mentioned (I found such a lack to be surprising given that the paper cites [15]). Some network features have been demonstrated to be redundant for the sake of NID classification (see [G, H, S]), and not mentioning them casts doubts on the experiments done in the paper. Note that I did not find any mentioning of these also in the supplementary material
* I was surprised of not finding [A] among the cited works, and especially among the "comparison" methods, given that it considers a very similar problem (and is also evaluated on the --fixed-- CIC-IDS17)
* Please report the size of the training / test data. I couldn't find a clear mention of this in the experimental section (note that this is a well-known problem in related literature [N]).
* The "bread and butter" of CL is that the performance should be measured "over-time". However, given the way the experiments are designed (at least based on how the results are presented), I cannot understand if this is truly the case.

The following is a list of paragraphs in the text for which I have concerns:

> NIDS must evolve continuously and effortlessly learn from the limited novel attack data.

What does "limited novel attack data" mean? How is such "novelty" determined?

> Furthermore, formulating NID as a supervised binary classification problem (SBCP) will be helpful in differentiating out-of-distribution normal samples from known intrusions well.

Isn't this the de-facto standard in many NIDS evaluations? I think this statement is redundant.

> (e.g., the difference between the number of samples of the minority classes DDOS attack-HOIC and SQL Injection of the CICIDS-2018 [18] dataset is 0.68 million).

Unclear.

> However, this strategy has a pitfall when the count of minority class samples in the finite buffer memory appears to come from a majority class.

Unclear. Plus, what does such "pitfall" lead to?

> making our approach suitable for large-scale training.

What does this mean? Plus, is this a problem in NID?

> Typically, the benign class samples must be chosen for replacement whenever new minority class samples arrive.

Define "typically". Is it "typical" in research? Is it typical "in practice"? On what grounds is this statement made?

> we use original class labels to organize the buffer memory to learn SBCP and chose class samples in memory with higher running statistic values for the replacement to accommodate newly arriving samples.

This statement is clear, but it contrasts with the underlying assumption of seeing NID as a binary classification problem. Indeed, providing (accurate) fine-grained attack labels is a tough problem in NID [N, R].

> This ensured that each task contained a mix of benign and attack data, maintaining the class imbalance resembling real-world network traffic.

How is the "resemblance" achieved?

> We created five tasks for KD249 DCUP’99 [22] and NSL-KDD [49], ten 250 for CICIDS-2017/2018 [18, 50], nine for UNSW-NB [24, 51, 52, 53, 54], and ten for ANOSHIFT [25] benchmark contains naturally occurring distribution shifts spanning over ten years of network traffic.

The last sentence does not connect with the previous ones.

> We use M=13333 for CICIDS-2017/2018, Anoshift, 5333/1333 for KDDCUP’99/NSL-KDD, 500 for SVHN, CIFAR-10/100 and 666/2666 for CLEAR-10/100.

I do not understand why the first settings have only "M", and the others have two numbers.

### EXTERNAL REFERENCES

[A]: Andresini, Giuseppina, et al. "INSOMNIA: towards concept-drift robustness in network intrusion detection." Proceedings of the 14th ACM workshop on artificial intelligence and security. 2021.

[B]: Apruzzese, Giovanni, et al. "The role of machine learning in cybersecurity." Digital Threats: Research and Practice 4.1 (2023): 1-38.

[C]: Kim, Daniel E., and Mikhail Gofman. "Comparison of shallow and deep neural networks for network intrusion detection." 2018 IEEE 8th Annual Computing and Communication Workshop and Conference (CCWC). IEEE, 2018.

[D]: Apruzzese, Giovanni, et al. "“Real Attackers Don't Compute Gradients”: Bridging the Gap Between Adversarial ML Research and Practice." 2023 IEEE Conference on Secure and Trustworthy Machine Learning (SaTML). IEEE, 2023.

[E]: Pontes, Camila FT, et al. "A new method for flow-based network intrusion detection using the inverse Potts model." IEEE Transactions on Network and Service Management 18.2 (2021): 1125-1136.

[F]: Apruzzese, Giovanni, Pavel Laskov, and Johannes Schneider. "SoK: Pragmatic Assessment of Machine Learning for Network Intrusion Detection." IEEE EuroS&P (2023).

[G]: Liu, Lisa, et al. "Error Prevalence in NIDS datasets: A Case Study on CIC-IDS-2017 and CSE-CIC-IDS-2018." 2022 IEEE Conference on Communications and Network Security (CNS). IEEE, 2022.

[H]: Engelen, Gints, Vera Rimmer, and Wouter Joosen. "Troubleshooting an intrusion detection dataset: the CICIDS2017 case study." 2021 IEEE Security and Privacy Workshops (SPW). IEEE, 2021.

[I]: Silva, João Vitor Valle, Martin Andreoni Lopez, and Diogo MF Mattos. "Attackers are not stealthy: Statistical analysis of the well-known and infamous kdd network security dataset." 2020 4th Conference on Cloud and Internet of Things (CIoT). IEEE, 2020.

[J]: Dietz, Christian, et al. "DMEF: Dynamic Malware Evaluation Framework." NOMS 2022-2022 IEEE/IFIP Network Operations and Management Symposium. IEEE, 2022.

[K]: Venturi, Andrea, et al. "Drelab-deep reinforcement learning adversarial botnet: A benchmark dataset for adversarial attacks against botnet intrusion detection systems." Data in Brief 34 (2021): 106631.

[L]: De Shon, Markus. "Information Security Analysis as Data Fusion." 2019 22th International Conference on Information Fusion (FUSION). IEEE, 2019.

[M]: Jacobs, Arthur S., et al. "AI/ML for Network Security: The Emperor has no Clothes." Proceedings of the 2022 ACM SIGSAC Conference on Computer and Communications Security. 2022.

[N]: Apruzzese, Giovanni, Pavel Laskov, and Aliya Tastemirova. "SoK: The impact of unlabelled data in cyberthreat detection." 2022 IEEE 7th European Symposium on Security and Privacy (EuroS&P). IEEE, 2022.

[O]: Sommer, Robin, and Vern Paxson. "Outside the closed world: On using machine learning for network intrusion detection." 2010 IEEE symposium on security and privacy. IEEE, 2010.

[P]: Hill, Mark D. "What is scalability?." ACM SIGARCH Computer Architecture News 18.4 (1990): 18-21.

[Q]: Luke, Edward A. "Defining and measuring scalability." Proceedings of Scalable Parallel Libraries Conference. IEEE, 1993.

[R]: Van Ede, Thijs, et al. "Deepcase: Semi-supervised contextual analysis of security events." 2022 IEEE Symposium on Security and Privacy (SP). IEEE, 2022.

[S]: Apruzzese, Giovanni, Luca Pajola, and Mauro Conti. "The cross-evaluation of machine learning-based network intrusion detection systems." IEEE Transactions on Network and Service Management (2022).

[T]: Pendlebury, Feargus, et al. "TESSERACT: Eliminating experimental bias in malware classification across space and time." Proceedings of the 28th USENIX Security Symposium. USENIX Association, 2019.


**Questions:**

I thank the authors for carrying out the research discussed in this paper. Ultimately, I **really** liked the paper: the research direction is crucial, the proposed methodology is "sensible", the evaluation is massive and (on the surface) rigorous. However -- as discussed in my review -- I have some doubts on the "practical" effectiveness of the proposed method.

Below is a set of questions that I invite the authors to answer. I must stress that my goal, here, is to assist the authors in realizing a work that is "outstanding": given the effort put by the authors in this submission, I have reason to believe that the authors are capable of this.

* Q1) What is the envisioned system infrastructure in which the proposed method is meant to be deployed?
* Q2) Can the authors, in the (short) timespan allotted for NeurIPS'23 review process, carry out additional experiments on a better dataset? (see my review)
* Q3) Can you explain Figure 3 in detail?
* Q4) What features are analyzed by the NIDS models?
* Q5) Please provide convincing arguments that the paper is not affected by data-snooping -- and, if this is not the case, then provide feasible ways in which the paper can be amended.
* Q6) Please elucidate how the "temporal" aspect is accounted for in the experiments. Suggestions are in [A, F, T].

Please answer these questions fairly.

**Limitations:**

The limitations are stated, but only in the supplementary material. However, according to this reviewer, these limitations should be expanded significantly (the reasons are provided in my review). For instance, most assumptions are overly optimistic (i.e., immutable attack landscape, widespread deployment of deep neural networks for NID) and the data hardly reflects a real network and/or which is in constant evolution.


## UPDATE AFTER AUTHOR's RESPONSE

I am increasing my score from a 5 to a 6, as well as the "Contribution" (from a 2 to a 3) in light of the clarifications and experiments carried out by the authors during the discussion phase.

---

> ### Author Rebuttal · Authors · 2023-08-09
>
> We sincerely request the reviewer ZBnA read the **responses to weakness section**  mentioned in the pdf (we could not specify here due to space limitations)  before reading our responses to questionnaires.
>
> **RESPONSES TO QUESTIONARIES**
>
> **R1**: The proposed mode does not require huge system infrastructure (memory and processing resources) as it uses a simple multi-layer perceptron (hardly with  5 to 6 hidden layers), which is a low complexity (capacity) architecture compared to large-size architecture like ResNet which has stacked convolution operations. So, it can also be deployed on edge devices.  Further, our solution is based on deep models, and we recommend using MLOps for low-latency inference in real-world deployments.
>
> **R2**:  To strengthen our findings, we carried out additional tests using CTU-13 [K], the newer CICIDS2017 and  CICIDS-2018 datasets ([G]). These results are presented in the Table2 of the PDF. Notably, our proposed methods consistently perform well on these datasets, aligning with the outcomes we previously shared in the paper. But, we could not run GSS-greedy and A-GEM because their competitiveness is equal  GEM method and also due to the short deadline.
>
> **R3)**  Memory replay-based continual learning methods, such as MIR, select specific samples from memory based on their impact on the model's parameter updates. They prefer samples with higher resulting losses. While these updates occur frequently during incoming data batches, they are only used for sample selection and not applied afterward. However, these frequent updates can introduce substantial computational overhead, especially for long data streams training deep models, due to the involvement of complex computations on large weight matrices. To address this, we investigated the relationship between regular and virtual updates of model parameters. Through t-SNE visualizations on datasets like CICIDS-2018, CIFAR-10, and CLEAR-10, we observed that virtual updates change gradually and tend to align with regular updates. To capture this phenomenon, we modified regular updates to approximate virtual ones. The challenge was quantifying this change. We approached this by examining the distribution of errors between virtual and regular updates across various model parameters, discovering a skewed Gaussian distribution. We used a two-component Gaussian mixture model (GMM) to represent this distribution, preserving individual error distributions for each parameter."
>
> **R4**: Here, we provide a list of features analyzed for each dataset.
>
> Regarding the newer CICIDS2017 and CICIDS-2018 datasets recommended by the reviewer, these datasets initially included more than 90 features. However, we removed flow-specific identifiers like flow ID, source and destination IPs, and timestamps, leaving approximately 51 features. Feature engineering was conducted using Pearson correlation with a threshold of 90%. Subsequently, the selected features underwent min-max normalization before training.
>
> Older CICIDS 2017 and CICIDS-2018 (used in prior experiments): These datasets initially contained nearly 80 features. Initially, we removed flow specific features. Then we excluded redundant features like Bwd PSH Flags, Bwd URG Flags,  and another six features as they contain invalid/zero entries and followed by normalizing data using a min-max normalizer. Then, we removed duplicate and inconsistent rows (similar feature values but with different labels). Eventually, we started the training process with the remaining 70 features.
>
> CTU-13: As recommended by this reviewer, we used the dataset from [K]. We perform min-max normalization on this dataset before training.
>
> KDDCUP & NSL-KDD:  For these datasets,  "service","is_host_login", and "num_outbound_cmds" features were removed as they were found to contain redundant values.
>
> USNW-NB15:  For this dataset, we removed 'srcip','sport','dstip','dsport','Stime','Ltime' features. We performed a categorical data encoding for the features like ‘proto’, ‘state’, and ‘service’, followed by min-max normalization, after which the training process was invoked with the remaining features.
>
> **R5)**: We have clarified potential data snooping in the previous section (Addressing Weakness). Here, we extend the discussion on the training process. The test set was strictly avoided throughout our training experiments. All experiments exclusively used training or validation sets, and data normalization relied on min-max techniques with the entire dataset, ensuring no test data influence on outcomes.
> Furthermore, we acknowledge the removal of timestamp features from certain datasets, which does not qualify as temporal snooping due to the formulation of the intrusion detection problem in continual learning. Our experiments operated in a domain learning setup, where consistent labels (0, 1) are assigned to each task, with label distributions evolving over subsequent tasks. This approach targets temporal distribution levels for each task, diverging from examining within-class (benign, attack) distribution using timestamps. We suggest employing RNN and LSTM alongside continual learning to capture temporal aspects within each class distribution.
>
> **R6)** The main theme of continual learning is to learn a set of tasks with varying distributions arriving sequentially without forgetting previously learned tasks (avoiding catastrophic forgetting).  In this paradigm, the temporal aspect is considered at the distribution level of each task (class incremental, task incremental, and domain incremental) rather than the temporal aspect within the distribution. Based on this intuition,  we conducted experiments in the domain incremental learning setting. As a result, we don't consider timestamps in some of our experiments. However, we advocate using RNN and LSTM with continual learning to capture temporal aspects within and between the distributions of different classes.

---

> > ### Comment · Reviewer_ZBnA · 2023-08-14
> > **Nice work**
> >
> > Dear authors,
> >
> > thank you for your response. I greatly appreciated the new experiments and the improved descriptions -- especially the ones focusing on RF.
> >
> > I still believe that the experimental settings excessively lean towards the "closed world" assumption, thereby undermining the "real value" of the proposed method. However, in light of the extensive and rigorous set of experiments (which has to be combined with the overall goal of the paper, which is commendable) I am confident that future research efforts can benefit from the research described in this work.
> >
> > For this reasons, I am increasing my score. However, I also endorse the authors to revise their paper by openly accounting for the "practical limitations" that affect their evaluation (in general, the paper needs a "tone down").

---

> > > ### Author Response · Authors · 2023-08-19
> > > **Thanking reviewer ZBnA**
> > >
> > > Dear reviewer **ZBnA**,
> > >
> > > We appreciate the amount of **time** and **effort** in providing the **most insightful** feedback on our submission. We agree to incorporate all your suggestions in the final version of the paper.

---

### Official Review · Reviewer_KabC · 2023-06-30

**Soundness:** 2 fair
**Presentation:** 2 fair
**Contribution:** 2 fair
**Rating:** 4
**Confidence:** 2

**Summary:**

The authors propose techniques for improving how to select samples for
replacement in the memory for continual learning and how to estimate
virtual SGD in MIR to reduce computation.

For replacement in memory, CBRS does not keep track of class counts,
and replacement might occur on non-majority samples in the memory.
Instead the authors propose ECBRS (Extended Class Balancing Reservoir
Sampling) keeps track of class counts (global information) and
replacement prefers classes with the highest count.  Each class c has
gamma(c) as the expected count in the memory with weights favoring
smaller classes.  Based on gamma(c), the next largest class might be
chosen.

They also propose Perturbation Assistance for Parameter Approximation
(PAPA) for MIR. In MIR sampling from the memory is informed by the
loss from virtual SGD parameter (VSP) updates for each sample, which
incur additional overhead.  They observe that VSP overlaps or scatter
around regular SGD parameter (RSP) updates.  They modeled the
difference between VSP and RSP with a two-component Gaussian Mixture
Model (GMM).  Then they estimate VSP from RST and GMM.  The GMM is
trained on one task and used in the remaining tasks.

Empirical results indicate that ECBRS generally outperforms 7 existing
on 12 datasets.  ECBRS can also improve the performance of MIR.  The
training time for PAPA is lower than MIR on 11 datasets, but achieve
similar accuracy.

I have read the authors' response and commented on them.

**Strengths:**

1.  The idea of estimaing virtual SGD updates is interesting.

2.  Empirical results indicate that ECBRS generally outperforms and
    PAPA can reduce training time.

**Weaknesses:**

1.  While PAPA is interesting, the reasoning for the error
    distribution from one task is applicable to another task could be
    further discussed/explored--see questions below.

2.  Font sizes for Table 2-4 and some figures are quite
    small--difficult to read.

3.  Some items can be clarified--see questions below.

**Questions:**

1.  Alg 1: "select a class that is the largest, having higher running
    statistics value and non-zero samples in the buffer. Otherwise,
    select a class with next higher running statistic value that has
    m_c >γ(c)."

    It seems the next highest class is selected when the largest class
    has no samples in the the buffer.  This seems to be different from:

    line 172: "once it reaches the threshold of the benign class
    (gamma(.)= 262), the ECBRS selects the class with
    the next highest running static value"

2.  line 233: The GMM is trained based on one task and is used for the
    other tasks.  This implies all the tasks has similar error
    distributions, which does not seem to be the case in Figure 3.  Learning
    a GMM for each task, and compare the GMMs would be helpful.

3.  Even if the error distributions are similar for the tasks in the paper,
     can other tasks have similar error distributions?  If not, a separate
     GMM would need to be learned, which might negate the computation
     savings in estimating VSP.

4.  line 266: "a randomly chosen single attack class and the remaining
    nine classified as benign."  Would one (large) benign class with
    the remaining (small) clases as attacks be more realistic (similar
    to the example in Figure 1)?.

Minor comment:

line 174: static -> statistic

**Limitations:**

Limitations of the proposed approach do not seem to be mentioned.

---

> ### Author Rebuttal · Authors · 2023-08-09
>
> **Q1: Alg 1: "select a class that is the largest, having higher running statistics value and non-zero samples in the buffer. Otherwise, select a class with next higher running statistic value that has m_c >γ(c)." It seems the next highest class is selected when the largest class has no samples in the the buffer. This seems to be different from:line 172: "once it reaches the threshold of the benign class (gamma(.)= 262), the ECBRS selects the class with the next highest running static value"**
>
> R1)Dear reviewer,  Thanks for raising this question.  We acknowledge that this issue caused due to the typo in the presented ECBRS algorithm, and it is required to add the condition m_c >= gamma(c) in line number 169 of the paper. This condition will ensure that irrespective of the higher running statistic value of the class ‘c’,  our proposed ECBRS method will not undermine beyond the threshold gamma(c).
>
> **Q3.Even if the error distributions are similar for the tasks in the paper, can other tasks have similar error distributions? If not, a separate GMM would need to be learned, which might negate the computation savings in estimating VSP.**
>
> R2:  As mentioned previously, incorporating a two-component Gaussian Mixture Model (GMM) has emerged as one of our key empirical findings. In this context, we aim to shed light on the fundamental nature of the two- GMM components from the perspective of task similarity. The computation of task similarity relies on the optimal transport dataset distance (OTDD).
>
> During this experiment, we design an experiment to learn tasks sequentially from MNIST, CIFAR-10. Before initiating the experiments, we calculate the mean OTDD value for each task, considering its relation to the remaining tasks. However, these values are not uniformly distributed compared to the experiments conducted solely using the CIFAR-10 dataset. This discrepancy suggests a higher dissimilarity within these experiments.
>
> |Dataset|Task1|Task2|Task3|Task4|Task5|Task6|Task7|Task8|Task9|Avg OTDD values acros all tasks(Mean±std)|
> |-|-|-|-|-|-|-|-|-|-|-|
> |CIFAR-10|258|255|220|243|212|241|225|236|244|237.2298 ±10.067|
> |MNIST+CIFAR10|310|**448**|310|**444**|310|**456**|310|**470**|310|374.179±**50.024**|
>
> Furthermore, during these experiments, we observe a significant disparity in the performance values between the MIR and PAPA algorithms. Notably, all these experiments are performed using 20 distinct task orders, and the reported results encompass both mean and standard deviation values.
>
> |Algorithm|PR-AUC(O)|PR-AUC(I)|ROC-AUC|
> |-|-|-|-|
> |MIR|**0.675±0.053**|**0.700±0.030**|**0.657 ±0.040**|
> |PAPA|0.645±0.047|0.661±0.099|0.628± 0.080 |
>
> Additionally, we aimed to validate this observation through a computational approach empirically. In this pursuit, after conducting experiments using the MIR algorithm, we calculated error values for the parameters and tried to fit the Gaussian Kernel Density Estimator (KDE). However, we encountered a singular matrix error while attempting the KDE fit, preventing us from generating a Gaussian approximation. It's noteworthy that this issue doesn't arise in other experimental scenarios. From this experience, we can deduce that the two-Gaussian Mixture Model is feasible whenever the variance in the mean OTDD value of each task concerning the remaining tasks is low.
>
> **Q2) The GMM is trained based on one task and is used for the other tasks. This implies all the tasks have similar error distributions, which does not seem to be the case in Figure 3. Learning a GMM for each task, and comparing the GMMs would be helpful**
>
> R2) We acknowledge the reviewer's concern about the error distribution.  In fig3, each row of subfigures refers to the error distribution of the model parameter on the particular datasets. This observation stands as a basis for developing the PAPA approach. To our knowledge, we assume stacking the error distribution figures of the different datasets might have brought ambiguity to the reviewer.  Once again, the error distribution for each dataset is displayed in a row labeled with the corresponding dataset name. Based on the response to question 3 (Q3), we advocate computing separate GMM for each task whenever task dissimilarity is high. For instance, in an experimental setting, like training jointly with MNIST and CIFAR-10 dataset.
>
>
> **Q4)Line 266: "a randomly chosen single attack class and the remaining nine classified as benign." Would one (large) benign class with the remaining (small) clases as attacks be more realistic (similar to the example in Figure 1)?.**
>
> **CIFAR-100** : Following your input, we conducted experiments on the CIFAR-100 dataset, focusing on its super-class labels. Six classes were grouped to form a larger benign task, then split into ten parts. Similarly, the remaining classes (shuffled together) were divided into fifteen parts, and we formed ten tasks. This approach aimed to maintain a class imbalance ratio of 1:5 or 1:20 in combined tasks. Remarkably, these results aligned with our main paper findings. Another notable finding was that the  ECBRS and PAPA consistently performed much better than all the other methods. ECBRS outperformed baselines by a large margin as the class imbalance ratio increased from 1:5 to 1:20.
>
>
> **CLEAR-10**: We also conducted experiments similar to CIFAR-100. Specifically, we used four classes across all the ten years to create a larger benign file and used the rest of the six classes to create 16 different attack classes in such a way that class imbalance per each task will be 1:20. Further, we split the larger benign class into ten parts and create nine tasks. In this experiment, each task will contain one benign and one attack class. Interestingly, even in this setup, the performance results stayed consistent with what we had reported in the main paper.
> **Due to the space limitation, we report the results of these experiments in Table 1 of the uploaded pdf**

---

> > ### Comment · Reviewer_KabC · 2023-08-17
> > **comments on response**
> >
> > Thanks for your response.
> >
> > Q2 and Q3:
> >
> > > we advocate computing separate GMM for each task whenever task dissimilarity is high
> >
> > You seem to have verified that error distributions could be different and separate GMMs would be needed.  This diminishes the results in the paper which are based on assuming the error distributions are similar and the same GMM is used with VSP across tasks/datasets.  Perhaps the paper can be improved by somehow estimating the similarity of error distributions among tasks (maybe via OTDD) and deciding if a current GMM can be reused with VSP or not.
> >
> > Q4:
> >
> > 1:5 or 1:20 imbalance ratio might still be small.  Depending what kinds of attack scenarios you are targeting, some scenarios would have significantly larger imbalance ratio.  For network intrusions, from the title of your paper, 1:100 is not uncommon.  Also, not sure why CIFAR-100 or MNIST is related to network intrusions.

---

> > > ### Author Response · Authors · 2023-08-19
> > > **Q2 and Q3: why the occurrence of task dissimilarity between two different tasks is rare in our setting?**
> > >
> > > **Response:**
> > > Here, we will demonstrate that the error distributions between different tasks remain similar even with varying task orders, and it can be modeled using two-component GMM. Additionally, we will illustrate cases where the dissimilarity in the error distribution occurs due to two dissimilar tasks is rare in the context of network intrusion detection systems (NIDS).
> > >
> > >
> > > **A two-component GMM is capable enough for various task orders**
> > >
> > > We direct the reviewer's attention to Section A.2.2 and Tables 5 and 6 of the supplementary material. In these sections, we conducted experiments using various datasets with four distinct task orders, each having a different first task. The performance results for each task order are presented in Table 6, and Table 5 illustrates the variance in these reported values. Notably, we observe that the variance of performance results falls within the range of $10^{-7}$ to $10^{-4}$. This observation indicates that the influence of the different first tasks used in the computation of the Gaussian Mixture Model (GMM) on the performance of PAPA is minimal. This underscores that the GMM constructed based on the initial error distribution (ED) effectively approximates the ED of subsequent tasks, irrespective of the choice of the first task.
> > >
> > > **Why the occurrence of two dissimilar tasks in a NIDS setting is rare?**
> > >
> > > We believe some clarification is needed here. Firstly, we want to clarify that the OTDD values reported in the table of the previous response related to the MNIST+CIFAR-10 experiment are not between two adjacent tasks. Instead, for a task (say t1), this value reported in the table is the mean OTDD value computed with other remaining tasks (t2,t3,...,t9) as follows.
> > >
> > > $$Mean.OTDD(t_{1}) =\frac{1}{8} \Sigma_{i=2}^{9} OTDD (t_{1},t_{i})$$
> > > We explicitly show these values to showcase an extreme case where the proposed PAPA method would not perform well. However, in practice finding these values is only possible if access to past and future tasks is allowed, but in the continual learning setting, such access is restricted. On the other hand, computing the OTDD values between two adjacent tasks will not help determine the higher similarity between the two tasks; refer to the table 1 below for an example.
> > >
> > >
> > > Table 1: OTDD values computed between two adjacent tasks
> > > | Experiment       | (0,1)  | (1,2)  | (2,3)  | (3,4)  | (4,5)  | (5,6)  | (6,7)  | (7,8)  | Mean± std     |
> > > | - | - | - | - | - | - | - | - | - | - |
> > > | MNIST + CIFAR-10 | 424.29 | 404.25 | 405.68 | 405.70 | 409.09 | 409.09 | 407.59 | 407.59 | 409.16 ±4.399 |
> > >
> > >
> > > The reason for calling the MNIST+CIFAR-10 experiment an extreme case is based on the following two observations.
> > > 1) In this experiment, training data arrives from two different data sources in the domain incremental setting with differing characteristics (size, image type, etc. refer to Table 2 below). However, in practice, finding a use case where the experiments with such a shift between tasks with different characteristics (switching between grayscale and RGB images learning sequentially) is difficult.
> > >
> > > Table2: characteristics of the MNIST and CIFAR-10 datasets
> > > |Dataset|Size|No of channels|Image type|
> > > | - | - | - | - |
> > > |MNIST|28 X 28|1|Grayscale|
> > > |CIFAR-10|32 X 32|3|RGB        |
> > >
> > > 2) The Anoshift dataset contains diverse traffic from five different networks [Ref1] of Kyoto University spanning ten years. Anoshift is the challenging dataset in network intrusion detection experiments, as also pointed out by the **reviewer ZBnA** (that contains data points from different networks) compared to other datasets (created over a short time on a single network, refer to Table 3 for mean OTDD values in each dataset, in which OTDD value is computed between adjacent tasks). For example, consider the datasets like CICIDS2017 and CICIDS2018, where datasets were framed using the traffic from a single network in which a high distribution shift may not happen. On the other hand, our proposed PAPA on the challenging Anoshift dataset (with diverse traffic) works better compared to baseline methods
> > >
> > >  Table 3: Mean OTDD values each task’s mean OTDD value computed with other remaining tasks
> > > | Dataset    | Mean OTTD values of all the tasks |
> > > | - | -|
> > > | CICIDS2017 | 0.1422 ±0.0439                    |
> > > | CICIDS2018 | 0.05635 ±0.0239                   |
> > > | ANoshift   | 5,611 ± 861.54                    |
> > >
> > >
> > >
> > > To conclude, our intention in framing MNIST+CIFAR-10 experiments is to showcase the extreme case where the proposed approach is expected to not perform well in domain incremental learning. However, encountering such a scenario  is rare in the NIDS setting.
> > >
> > > [Ref1]. Song, J., Takakura, H., Okabe, Y., Eto, M., Inoue, D., & Nakao, K. (2011, April). Statistical analysis of honeypot data and building of Kyoto 2006+ dataset for NIDS evaluation. In Proceedings of the first workshop on building analysis datasets and gathering experience returns for security (pp. 29-36).

---

> > > > ### Author Response · Authors · 2023-08-19
> > > > **Q4. Experimenting with an imbalance ratio of 1:100**
> > > >
> > > > **Response**:: In our research, we view intrusion detection as a sequence of distribution shifts (DS) in each task, encompassing benign and attack data in a supervised learning context. We measure these shifts using the optimal transport dataset distance (OTDD). While most Network Intrusion Detection Systems (NIDS) exhibit minimal distribution shifts, ANOSHIFT is an exception. Reduced OTDD values indicate lesser distribution shifts. To comprehensively assess our approach, we employed image datasets with substantial distribution shifts, meticulously emulating the creation of IDS tasks. Refer to Table 4 for OTDD values of each dataset.
> > > >
> > > > Table 4:  Average OTDD values of each dataset, in which OTDD value is computed between adjacent tasks
> > > > | Dataset    | Avg. OTDD values across all the tasks |
> > > > | ---------- | ------------------------------------- |
> > > > | CICIDS2018 | 0.0508                                |
> > > > | CICIDS2017 | 0.1527                                |
> > > > | CTU-13     | 4.44                                  |
> > > > | CIFAR-10   | 231                                   |
> > > > | CIFAR-100  | 3139                                  |
> > > > | ANoshift   | 7189                                  |
> > > >
> > > >
> > > > The performance results on two representative datasets (CIFAR-100 and CLEAR-10) are reported in Table 5 and Table 6. CIFAR-100 contains distribution shifts due to the **synthetic** formulation of the tasks, whereas CLEAR-10 has **natural** distribution shifts between the tasks as their data spans over a decade. We observe that the proposed methods either outperform or are on par with the baseline methods, following a similar trend to the previously reported results.
> > > >
> > > >
> > > >
> > > > **Table 5: Performance results on the CIFAR-100 dataset with an imbalance ratio of 1:100.**
> > > > | Methos     | PR-AUC (O)     | PR-AUC (I)        | ROC-AUC           | Train time     |
> > > > | ---------- | -------------- | ----------------- | ----------------- | -------------- |
> > > > | EWC        | 0.644 ±  0.014 |     0.603±  0.013 | 0.644±  0.013     | 1197           |
> > > > | GEM        | 0.653 ±  0.020 | 0.626 ±  0.026    | 0.643    ±  0.015 | 772            |
> > > > | AGEM       | 0.638 ±  0.022 |   0.591 ±  0.014  | 0.638±  0.016     | 558            |
> > > > | GSS-greedy | 0.659 ±  0.024 | 0.646 ±  0.025    | 0.662±  0.019     | 4837           |
> > > > | SI         | 0.645±  0.008  | 0.614 ±  0.018    |   0.635±  0.011   | 283            |
> > > > | CBRS       | 0.572 ±  0.012 | 0.598 ±  0.034    | 0.572±  0.021     | 72             |
> > > > | MIR        | 0.629±  0.021  | 0.606 ±  0.064    | 0.611 ±  0.045    | 111            |
> > > > | ECBRS      | 0.663 ±  0.013 | 0.611 ±  0.014    | 0.663 ±  0.010    | 74             |
> > > > | MIR+ECBRS  | 0.679±  0.029  | 0.665 ±  0.045    | 0.671±  0.035     | 118            |
> > > > | PAPA       | **0.680±  0.041**  | **0.668±  0.038**     | **0.672 ±  0.036**    | 77 (**SE = 34 %**) |
> > > >
> > > >
> > > >
> > > > **Table 6: Performance results on CLEAR-10  dataset with imbalance ratio 1:100**
> > > > | Methos     | PR-AUC (O)      | PR-AUC (I)     | ROC-AUC         | Training time  |
> > > > | ---------- | --------------- | -------------- | --------------- | -------------- |
> > > > | EWC        | 0.858 ±  0.011  | 0.828±  0.015  | 0.835 ±  0.011  | 421            |
> > > > | GEM        | 0.858 ±  0.015  | 0.852 ±  0.015 | 0.848±  0.007   | 965            |
> > > > | AGEM       | 0.853 ±  0.014  | 0.822 ±  0.025 | 0.833 ±  0.014  | 645            |
> > > > | GSS-greedy | 0.881 ± 0.011   |   0.85±  .007  |   0.861± 0.010  | 1456           |
> > > > | SI         | 0.847±  0.013   | 0.833±  0.006  | 0.835 ±  0.008  | 204            |
> > > > | CBRS       | 0.941 ±  0.009  | 0.927 ±  0.009 | 0.931±  0.009   | 613            |
> > > > | MIR        | 0.886 ±  0.010  |   0.89±  0.007 |   0.883±  0.006 | 321            |
> > > > | ECBRS      | 0.937 ±  0.005  | 0.926±  0.007  |   0.926±  0.006 | 534            |
> > > > | MIR+ECBRS  | **0.955  ±  0.005** | **0.94 ±  0.008**  | **0.946±  0.004**   | 371            |
> > > > | PAPA       | 0.952 ±  0.004  | 0.933±  0.006  |   0.941±  0.005 | 172 (SE = **53%**) |
> > > >
> > > >
> > > >
> > > > **On Limitations of the proposed approach**
> > > >
> > > > Eventually, we direct the reviewer to section **A.8** of the supplementary material, where we have addressed the limitations of the proposed methods to the best of our knowledge.

---

> > > > > ### Comment · Reviewer_KabC · 2023-08-19
> > > > > **comments on response**
> > > > >
> > > > > Thanks for the response.
> > > > >
> > > > > Based on the network intrusion datasets that were studied, the error distributions remain similar across tasks.  I suggest including some measurements (similar to those in your response) in the (main) paper to illustrate that observation--it could be one plot: x-axis is a pair of adjacent task numbers, y-axis is the measurement, and different curves for different datasets.  That is, the plot helps justify the reuse of GMM with VSP.
> > > > >
> > > > > Since the title of your paper states network intrusion, I suggest including only datasets that are network intrusion in terms of evaluation.  That is, focus on network intrusion.  Evaluations on other (non network intrusion) datasets could be in the supplementary.
> > > > >
> > > > > I'm going to increase my score by one level.

---

> > > > > > ### Author Response · Authors · 2023-08-20
> > > > > > **Thanks to Reviewer KabC**
> > > > > >
> > > > > > We thank Reviewer KabC for the suggestions to improve our exposition, as well as the score increase. We will incorporate the suggestions in the main paper.

---

> ### Comment · Area_Chair_FFZB · 2023-08-16
> **Check the rebuttal**
>
> @Reviewer KabC,
>
> Does the rebuttal address your concerns?

---

### Official Review · Reviewer_wTtL · 2023-07-05

**Soundness:** 3 good
**Presentation:** 3 good
**Contribution:** 3 good
**Rating:** 6
**Confidence:** 4

**Summary:**

This paper improves upon existing memory replay-based continual learning methods for anomaly detection. First the authors extend class balancing reservoir sampling (CBRS) and develop ECBRS, using global information in order to keep more accurate information about class imbalance. Second, the authors proposed a perturbation-assisted parameter approximation (PAPA) method for estimating virtual SGD parameter (VSP) updates, resulting in reduced training time for methods like maximally interfered retrieval (MIR). The proposed methods are evaluated on intrusion detection, computer vision, and anomaly detection datasets, and shown to increase performance while reducing training time.

**Strengths:**

This paper relies on simple heuristics that are shown to be quite effective for improving upon existing CL methods. The authors have also done extensive evaluation of their methods using several datasets, including ablation studies. In addition to performance improvements, the proposed methods can also reduce training times, resulting in better scalability.

**Weaknesses:**

- Since the paper heavily relies on CBRS and MIR, I suggest that the authors briefly describe these methods to make the paper more standalone and to make its novelties more clear.
- While the proposed methods are sound, I am not sure if the utilized evaluation datasets can fully support the paper's claims. The paper mainly focuses on learning from data with distribution shifts, however all but one of the evaluated datasets correspond to a short time period with little to no distribution shift.
- Related to the above, I am not sure if the inclusion of image datasets is needed as the proposed methods are presented for network intrusion datasets. The authors mention that they include image dataset to evaluate learning on data with distribution shifts, but I am not sure if image datasets contain any distribution shift.
- The authors should also include the standard deviation of AUC scores in Tables 2, 3.

**Questions:**

- Do the authors have an idea on why PAPA achieves better performance as compared to MIR in Table 3? Since PAPA performs an approximation, I would have expected the AUC scores to be slightly lower than those for MIR.
- Do the authors have an insight as to why the error between regular/virtual parameter updates resembles a two-component GMM? Is it reasonable to expect this assumption to hold on other datasets?
- What is the reason for including image datasets for evaluation. If the reason is distribution shifts, can you state why you expect these datasets to contain distribution shifts?

**Limitations:**

Limitations are adequately discussed.

---

> ### Author Rebuttal · Authors · 2023-08-09
>
> **Q1) Do the authors have an idea on why PAPA achieves better performance as compared to MIR in Table 3? Since PAPA performs an approximation, I would have expected the AUC scores to be slightly lower than those for MIR**.
>
> R1). We sincerely thank the reviewer for bringing up this important observation. After your feedback, we thoroughly investigated the issue and successfully identified the root cause. The problem arose from using **drop_last=False** in the data loader for specific datasets. We have addressed this issue and re-conducted the experiments. As a result of the fix, we now observe consistent and uniform behavior across all the datasets. The affected datasets, namely NSL-KDD, CICIDS2017, ANOSHIFT, and CIFAR-100, have all been rectified. Additionally, we noticed a minor discrepancy in the results of SVHN. However, this discrepancy is insignificant, as it only appears at the third decimal position and can be attributed to noise in the training process, which we can safely ignore. (refer to table below). We appreciate your diligence in reviewing our work, and your feedback has helped us improve the accuracy and reliability of the results presented in the paper.
> |Dataset|Method|PR-AUC(O)|PR-AUC (I)|ROC-AUC| Train time |
> |-|-|-|-|-|-|
> |NSL-KDD|MIR+ECBRS|0.964±0.009|0.971±0.003|0.97±0.004|27.86|
> ||PAPA|0.961±0.017|0.968±0.007|0.969±0.006|20.75|
> |CICIDS2017|MIR+ECBRS|0.994±0.001|0.993±0.000|0.993±0.000|398.4|
> ||PAPA|0.994±0.001|0.992±0.003|0.993± 0.003| 239.8|
> |ANOSHIFT|MIR+ECBRS|0.944±0.004|0.926±0.011|0.934±0.009|1273.4|
> ||PAPA|0.947±0.005| 0.927±0.011| 0.934±0.007|982.2|
> |CIFAR-100|MIR+ECBRS|0.882±0.011|0.864±0.014| 0.871±0.013| 227.4|
> ||PAPA|0.877±0.011|0.861±0.012|0.870±0.010|119.8|
>
>
> **Q2.Do the authors have an insight as to why the error between regular/virtual parameter updates resembles a two-component GMM? Is it reasonable to expect this assumption to hold on other datasets?**
>
> R2:  As mentioned previously, incorporating a two-component Gaussian Mixture Model (GMM) has emerged as one of our key empirical findings. In this context, we aimed to shed light on the fundamental nature of the two GMM components from the perspective of task similarity. The computation of task similarity relies on the optimal transport dataset distance (OTDD)[2].
>
> We designed an experiment to learn tasks sequentially from MNIST and CIFAR-10 during this study. Before initiating the experiments, we calculated each task's mean OTDD[2] value, considering its relation to the remaining tasks. However, these values are not uniformly distributed compared to the experiments conducted solely using the CIFAR-10 dataset. This discrepancy suggests a higher dissimilarity within these experiments.
>
> | Dataset | Task1 | Task2 | Task3 | Task4 | Task5 | Task6 | Task7 | Task8 | Task9 |  Avg OTDD value across all tasks(Mean±std )|
> |-|-|-|-|-|-|-|-|-|-|-|
> | CIFAR-10| 258| 255| 220| 243| 212| 241| 225|236| 244| 237.2298±10.067|
> | MNIST+CIFAR10 | 310| **448**|310|**444**|310|**456**|310|**470**|310|374.179±**50.024**|
>
> Furthermore, during these experiments, we observed a significant disparity in the performance values between the MIR and PAPA algorithms. Notably, all these experiments are performed using 20 distinct task orders, and the reported results encompass both mean and standard deviation values.
> | Algorithm | PR-AUC (O)| PR-AUC (I)| ROC-AUC|
> |-|-|-|-|
> | MIR| **0.675 ± 0.053** |**0.700  ± 0.030** | **0.657 ±0.040**|
> | PAPA| 0.645 ± 0.047| 0.661   ± 0.099 | 0.628± 0.080|
>
> Additionally, we aimed to validate this observation through a computational approach empirically. In this pursuit, after conducting experiments using the MIR algorithm, we calculated error values for the parameters and endeavored to fit the Gaussian Kernel Density Estimator (KDE). However, we encountered a singular matrix error while attempting the KDE fit, preventing us from generating a Gaussian approximation. It's noteworthy that this issue doesn't arise in other experimental scenarios.
>
> From this experience, we can deduce that the two-Gaussian Mixture Model is feasible whenever the variance in the mean OTDD value of each task concerning the remaining tasks is low.
>
> **Q3: What is the reason for including image datasets for evaluation? If the reason is distribution shifts, can you state why you expect these datasets to contain distribution shifts?**
>
> R3:  ANOSHIFT captures long-term distribution shifts (DS), but many NID benchmarks lack concept drift [1]. Without relevant cybersecurity benchmarks, the proposed methods were tested on computer vision benchmarks and modified to mimic the behavior of the IDS benchmark. The ease of visualizing distribution shifts makes CV benchmarks suitable for validating the proposed approaches. These DS in image classification can also be quantified using optimal transport dataset distance (OTDD)[2] computed between two tasks, where higher OTDD values indicate greater distribution shifts. For instance, on CIFAR-10 and CIFAR-100 datasets, mean OTDD values are 205 and 3139, respectively
>
> |Dataset| Avg.OTDD values across all the tasks|
> |-|-|
> |CICIDS2018|0.0508|
> |CICIDS2017| 0.1527|
> |CTU-13|4.44|
> |CIFAR-10|231|
> |CIFAR-100|3139|
> | ANoshift|7189|
>
> [1] Dragoi, M., Burceanu, E., Haller, E., Manolache, A., & Brad, F. (2022). AnoShift: A distribution shift benchmark for unsupervised anomaly detection. Advances in NIPS, 35, 32854-32867.
>
> [2]Alvarez-Melis, D., & Fusi, N. (2020). Geometric dataset distances via optimal transport. Advances in NIPS, 33, 21428-21439.
>
>
> **Weakness**
>
> *Please read R1 as a response to weakness one identified by the reviewer*
>
> **R1**) We will add more description about CBRS in supplementary material
>
> **R4**) We observe that the std for most experiments across all the datasets falls in the (0.001 to 0.010) range, implying low variance. So, we exclude them in the main paper.  Considering your feedback, we agree to add them to the final paper.

---

> > ### Comment · Reviewer_wTtL · 2023-08-19
> >
> > Thank you for your response and the additional tests. I especially appreciate investigating and addressing my first question and agreeing to add more details about CBRS.

---

> > > ### Author Response · Authors · 2023-08-19
> > > **Thanking reviewer wTtL**
> > >
> > > Dear reviewer **wTtL**,
> > >
> > > We thank you for providing **constructive feedback**,  especially the **AUC comparison** between the MIR and PAPA. This helped in maintaining the accuracy and reliability of the results. We agree to incorporate all your suggestions in the final version of this paper.

---

### Author Rebuttal · Authors · 2023-08-09

This pdf contains a table1 representing the results corresponding to the question (Q4) raised by the reviewer **KabC**.  Specifically, table results are obtained on CIFAR-100 and CLEAR-10 datasets.  Another table (table2) contained the results on newer datasets like CTU-13, modified CICIDS-2017 and CICIDS-2018 as suggested by the reviewer  **ZBnA**. Additionally, we also include our responses related to  weakness identified by the reviewer **ZBnA**

---

### Decision · Program_Chairs · 2023-09-21

**Decision:**

Accept (poster)

**Comment:**

The paper proposes two novel methods to improve continual learning for network intrusion detection with class imbalance and scalability challenges, with experiments showing performance improvements on various benchmarks.

Two reviewers are positive, and one reviewer is negative, whose main concern (regarding why the error distribution from one task is applicable to another task) was largely addressed during the discussion period.

Overall, it is a good work.